# Nanoparticles in Medicine: Current Status in Cancer Treatment

**DOI:** 10.3390/ijms241612827

**Published:** 2023-08-15

**Authors:** Krešimir Pavelić, Sandra Kraljević Pavelić, Aleksandar Bulog, Andrea Agaj, Barbara Rojnić, Miroslav Čolić, Dragan Trivanović

**Affiliations:** 1Faculty of Medicine, Juraj Dobrila University of Pula, Zagrebačka 30, 52100 Pula, Croatia; 2Faculty of Health Studies, University of Rijeka, Ulica Viktora Cara Emina 5, 51000 Rijeka, Croatia; 3Teaching Institute for Public Health of Primorsko-Goranska County, Krešimirova Ulica 52, 51000 Rijeka, Croatia; 4Faculty of Medicine, University of Rijeka, Braće Branchetta 20, 51000 Rijeka, Croatia; 5Clear Water Technology Inc., 13008 S Western Avenue, Gardena, CA 90429, USA; mcolic@cwt-global.com; 6Department of Oncology and Hematology, General Hospital Pula, Santorijeva 24a, 52200 Pula, Croatia

**Keywords:** nanoparticles, cancer treatment, metallic nanoparticles, quantum dots, polymeric nanoparticles, carbon nanotubes, graphene

## Abstract

Cancer is still a leading cause of deaths worldwide, especially due to those cases diagnosed at late stages with metastases that are still considered untreatable and are managed in such a way that a lengthy chronic state is achieved. Nanotechnology has been acknowledged as one possible solution to improve existing cancer treatments, but also as an innovative approach to developing new therapeutic solutions that will lower systemic toxicity and increase targeted action on tumors and metastatic tumor cells. In particular, the nanoparticles studied in the context of cancer treatment include organic and inorganic particles whose role may often be expanded into diagnostic applications. Some of the best studied nanoparticles include metallic gold and silver nanoparticles, quantum dots, polymeric nanoparticles, carbon nanotubes and graphene, with diverse mechanisms of action such as, for example, the increased induction of reactive oxygen species, increased cellular uptake and functionalization properties for improved targeted delivery. Recently, novel nanoparticles for improved cancer cell targeting also include nanobubbles, which have already demonstrated increased localization of anticancer molecules in tumor tissues. In this review, we will accordingly present and discuss state-of-the-art nanoparticles and nano-formulations for cancer treatment and limitations for their application in a clinical setting.

## 1. Introduction

Cancer remains among the leading causes of death worldwide, with 4.5 million (29.8%) deaths attributed to cancer according to the World Cancer Reports [1], which is mainly due to cases diagnosed at late stages or metastases. The main cancer treatments include surgery, radiotherapy, chemotherapy, hormone therapy, immunotherapy or combinations of these therapies [2]. Some of these treatment modalities, such as, for example, chemotherapy, lack specificity and have problems with cytotoxicity, stem-like cell growth and multidrug resistance [3,4,5]. When we talk about the treatment of malignant diseases, highly cytotoxic non-selective compounds, as well as targeted smart drugs, are aimed at individual molecular targets and often have a large number of side effects, do not contribute to a significant increase in the survival of patients, especially for those diagnosed with later stages of the disease with metastases, and are very expensive [3,4,5].

Nanotechnology has been acknowledged as one possible solution to improving existing cancer treatments or to offering innovative therapeutic solutions. Through the use of nanoparticles (NPs), indeed, some important goals in cancer therapy may be achieved: decreased adverse effects of delivered drugs, the possibility to prepare a myriad of nano-formulations for drug delivery, and the targeting of tumoral cells and their destruction with the help of NPs’ electrical, magnetic or optical characteristics [6,7,8]. Indeed, the property of nanosized materials makes them accumulate in tumor tissues due to leaky vasculature and underdeveloped lymphatic drainage. These specific changes in the tumor vasculature, termed the enhanced permeability and retention effect (EPR), had been discovered decades ago by Maeda and Matsumura [6] and have been extensively studied for delivery of various NPs and applications in cancer treatment hitherto [7,8]. In particular, in tumor tissue, the endothelium of blood vessels becomes more permeable than in a normal, healthy state. This happens because of hypoxia, where quickly growing tumors recruit new vessels or overflow the existing ones, and those vessels are leaky and ensure passive transport of a wider range of small particles entering the cell. Due to the misfunction of normal lymphatic drainage, NPs can stay inside the tumor and its tissue much longer, while small molecule drugs, which are known to have a short circulation time, are washed out first. Here, it should be stated that, since its discovery up to now, the EPR effect was found not to be universal to all tumors and, especially, it is less observed in large tumors due to large tumor mass obstructing the blood flow and vasculature. In addition, targeting of the tumor by nanoparticles also requires stable particles that have an increased plasma half-life [9]. These controversies regarding EPR as a tumor-specific property are additionally corroborated by the facts that EPR-dependent drug delivery is affected not only by tumor heterogeneity but is more pronounced in animal tumor models used to study EPR than in human patients [10]. Finally, tumor targeting should be clearly differentiated from cell targeting, the latter being very specific due to the use of chosen antigen targeting, i.e., cell receptors, such as, for example, EGFR (epidermal growth factor receptor) [11], while targeting of tumor tissue, i.e., the stroma or tumor microenvironment, relies on targeting of histological, physiological or biochemical properties of tumors [12,13]. A specific problem in NPs’ delivery to tumors is the presence of biological barriers, such as the tumor microenvironment (TME), i.e., in pancreatic ductal adenocarcinoma, or the blood–brain barrier (BBB) in brain tumors. The TME is a biologically functional barrier based on a system composed of abnormal vasculature, fibroblasts and various immune cells, all fixed in an extracellular matrix and with a pressure gradient between the interstitial space and a tumor mass. Tumor cells in the core become hypoxic and the anoxic metabolic pathway leads to a decrease in pH status [14,15]. Strategies to reduce interstitial fluid pressure to improve tumor penetration include extracellular matrix targeting, drug interventions to normalize the vasculature, applications of hypertonic solutions to shrink extracellular cells, hyperthermia, radiofrequency and high-intensity focused ultrasound to enhance the delivery of nano-sized drugs. Having a different mechanism, but also being a difficult barrier to pass through, the BBB is a multicomplex barrier in the central nervous system which prevents substances from entering the brain. It is interesting that, while metastases outside the central nerve system (CNS) often pass through the BBB into the CNS, primary brain tumors almost never pass in the opposite direction. This mechanism has not yet been investigated, but it could be used to enable the passage of nanoparticles through this barrier in both directions. There have been tremendous efforts in overcoming the BBB for drug delivery in general, and features of nanoparticles make nano-carriers appealing for research and design. Straehla et al. [16] developed drug-carrying nanoparticles using a human tissue model which accurately replicated the BBB and showed that the particles could enter brain tumors and induce glioblastoma cell death.

NPs may generally be used to improve the pharmacokinetics and biodistribution properties of drugs [17]. The NPs studied in the context of cancer treatment include both organic and inorganic particles, whose roles may often be expanded into diagnostic applications. According to our literature search, some of the best studied NPs include metallic gold and silver NPs, quantum dots (QDs), polymeric NPs, carbon nanotubes and graphene. Recently, novel NPs were developed and studied for improved cancer cell targeting, such as, for example, nanobubbles, which have already demonstrated an increased localization of anticancer molecules in tumor tissues [18]. In this review, we accordingly present and discuss the state-of-the-art NPs and nano-formulations for cancer treatment and the current limitations for their application in a clinical setting.

## 2. Types of Nanoparticles

### 2.1. Inorganic Nanoparticles

Inorganic NPs are generally biocompatible and highly stable compared to organic materials. These NPs include quantum dots, metallic nanoparticles, polymers and porous nanomaterials, the latter mainly based on silica materials. Usually, inorganic NPs comprise an inorganic core and an organic shell for biocompatibility and functionalization in targeted drug delivery [19]. In Table 1 and Table 2, the approved inorganic nanomedicine drugs on the market for cancer treatment and the inorganic nanomedicine drugs under clinical trial for cancer treatment are presented based on the most recent study, carried out in the year of 2023 by Nirmala et al. [20].

#### 2.1.1. Quantum Dots (QDs)

Quantum dots (QDs) are inorganic materials that have semiconducting properties and are, usually, synthesized with the II–VI or III–V elements of the periodic table. Their size is generally from 10 to 100 Å in radius. QDs have specific physicochemical properties that result from a combination of their crystalline metalloid core structure/composition and quantum-size confinement, which occurs when metal and semiconductor particles (QD cores) are smaller than their Bohr radii (~1–5 nm), which allows them to have semiconductor properties [27]. Recently, the most common use of QDs has been for in vitro bioimaging, due to their properties of narrow emission, wide-range UV excitation, high photostability and bright fluorescence [28]. QDs, however, are potentially toxic due to their properties that lead to the release of heavy metal ions in vivo, causing genotoxicity, particularly in the case of the first-generation QDs containing cadmium [29], as well as organelle dysfunction [30] and oxidative damage [31]. Accordingly, QDs that contain silicon (Si-QDs), carbon (C-QDs), graphene (GQDs), silver (Ag_2_Se), silver and sulfur (Ag_2_S) or copper and zinc (CuInS2/ZnS), have been increasingly studied [32]. In addition, QD surface modification by the use of biocompatible molecules is an alternative approach studied in recent years to lower their toxicity in vivo. For example, QDs coated with polymer/silica [33], silica particles [34] or poly-thiol [35] have been prepared for in vivo applications. In general, QDs generate intracellular reactive oxygen species (ROS), thereby causing cancer cell death through oxidative DNA damage [36], or directly impacting the immunological processes through the enhancement of proinflammatory signaling through different immune modulators, including cytokines, chemokines and metalloproteinases [37]. This is why in vivo applications of QDs have an interesting potential for cancer therapy, either by the use of QDs alone or using QDs conjugated with anticancer drugs [38]. Still, an overaccentuated proinflammatory effect induced by QDs may be dangerous for the patient and more studies will be required in this area to find the appropriate QD and a corresponding dosage for a desired effect without harmful effects on the organism in vivo. Indeed, the majority of mechanistic studies are currently available from in vitro experiments and cannot be directly correlated with in vivo effects. One such QD-based system containing doxorubicin was, for example, tested on tumor cell lines in vitro and exerted an increased cytotoxicity and genotoxicity as a folate-receptor through interactions with the C-rich regions of these genes’ promoters in a study by Luo et al. on a doxorubicin targeted delivery system for A549 lung cancer cells [39]. Sun et al. studied the use of QDs in overcoming the multidrug resistance phenomena (MDR) in vitro on an A549 lung cancer cell line. In particular, they observed significant inhibition of P-glycoprotein (P-gp) expression in cells treated with P-gp-miR-34b and P-gp-miR-185 conjugated with CdSe/ZnS-MPA QDs and CdSe/ZnS-GSH QDs. Indeed, the choice of miRNAs in that study was due to the previously observed correlation of their decreased expression with increased P-gp activity and cancer drug resistance in tumor cells, an unwanted cancer therapy outcome that often hampers cancer treatment [40]. Similarly, graphene QDs also lowered the expression of P-glycoprotein, multidrug resistance protein MRP1 and breast cancer resistance protein genes-resistant MCF-7 cells [41]. QDs, indeed, have potential in connection with MDR phenomena and they were tested with promising results also as blockers of ABC efflux transporters in human breast cancer SK-BR-3 cells [42], as P-gp protein expression down-regulators and apoptosis inducers in lymphoblastoid cells and BALB/c nude mice [43], and as inhibitors of P-gp drug efflux pumps, which was accompanied by overexpression of apoptosis-related caspase proteins in human hepatocarcinoma cells and nude mice [43].

#### 2.1.2. Metallic Nanoparticles

Metallic nanoparticles with a usual diameter of 1–100 nm (metallic NPs) are often metal oxides or a metallic core structure coated with an organic material. These particles bear interesting physical and chemical properties that may be exploited in cancer therapy. Some of the advantages of such systems are easy synthesis, a functional surface that may increase their affinity and selectivity to target molecules, a large surface-area-to-volume ratio and magnetic properties for some iron-based particles [44]. The surface of the metallic NPs can be easily adjusted to interact with targeting agents and other molecules through H-bonds, covalent bonds and electrostatic interactions [45]. Importantly, the metallic NPs exhibit increased stability and half-life in circulation, biodistribution and specific targeting into the target site, which is relevant for their clinical application. These properties are due to surface modification of NPs with thiol group, disulfide ligands, amines, nitriles, carboxylic acids, and phosphines or polyethylene glycol (PEG) [46]. Metallic NPs can be used to direct therapeutic agents to the desired site of action, minimizing the drug effects on healthy tissues and controlling the release of the drug [47,48]. In addition, metallic NPs conjugated with drugs may directly target cell surface receptors such as, for example, the HER2 (human epidermal growth factor receptor 2) receptor in breast tumor cells [49]. Some of the metallic NPs tested in clinical applications include silver (AgNPs), gold (AuNPs), palladium, titanium, zinc, and copper-based NPs [46]. Clinical applications stemmed from in vitro data on metallic NP mechanisms of action. As an example, Barabadi et al. reviewed all the published data for AgNPs’ effects on in vitro lung cancer cell lines, in the context of their potential usage in the fight against lung cancer, and they have, for example, found a potential of AgNP in reversal of cancer MDR. The main AgNP-induced cytotoxicity mechanisms covered by that study included the induction of excessive production of ROS, an increase in apoptotic enzyme levels and apoptosis [50]. In addition, AgNPs have been tested on leukemia cell lines [51], on breast tumor cells [52], and on hepatocellular and colon cancer cell lines [53] in vitro, where they showed antiproliferative and cytotoxic effects due to several mechanisms identified by the studies’ authors, including the induction of ROS, mitochondrial damage and the induction of apoptosis. AuNPs have also been tested as cancer treatment options and have been prepared in different shapes and forms, including gold nanospheres, gold nanorods and gold nanocages [44]. For example, an AuNP drug carrier system for doxorubicin delivery, functionalized with PEG polymers and polyamido-amine (PAMAM) dendrimers, has been developed as a potential intracellular delivery vehicle and tested on A549 cells in vitro [54]. The approved inorganic nanomedicine drugs for cancer treatment and the inorganic nanomedicine drugs under clinical trial for cancer treatment are presented in Table 1 and Table 2, respectively.

#### 2.1.3. Inorganic Porous Nanomaterials

Inorganic porous nanomaterials have been acknowledged as promising drug carriers due to particular structural properties, such as high loading capacity of biomolecules, possibilities of various surface modifications and controllable release of drug molecules [55]. These are due to their porous solid structure, with particle units forming uniform pore structures of microporous and mesoporous range [56]. The majority of porous materials are made of randomly oriented and repeating single units forming pores [57]. Pores are thus used for the embedding of desired molecules, i.e., drugs or antibodies. Especially important are porous materials where the channels are unidirectionally oriented over a macroscopic scale so that the fate of embedded molecules in the channels can be better controlled [57]. A large group of porous materials are zeolites, crystalline aluminosilicates with a porous structure on the micro- and nano-scale that occur naturally but can also be prepared in controlled reactions to obtain materials with controlled physico-chemical properties and pore size [58]. The single unit, the building block of a zeolite material, is a tetrahedron made of atoms (such as Si and Al) that are bound together through oxygen atoms in between two tetrahedron units [58]. Such linked tetrahedra form channels and pores, often containing water and alkali metals that stabilize the zeolite structure, resulting in stable zeolite structures that act as molecular sieves, adsorbing materials and ion-exchangers [58]. Zeolites have, accordingly, been recently prepared as nanosized materials to advance their usage in industrial applications [59]. Their potential uses in medicine and therapy, however, are among the most promising. In particular, this is because zeolite nanocrystals are stable in colloidal suspensions, and different drugs, or therapeutic and diagnostic agents, can be placed in their pores or cages or attached to their surfaces [60]. A recent study, for example, showed a high rate of internalization and intracellular localization of a nanosized zeolite X (faujasite-X) into glioblastoma cells, especially in hypoxia circumstances, and without showing cytotoxic effects. The authors proposed these nanozeolite crystals, accordingly, as carriers for drugs to target cancer cells [61]. So far, several zeolites of meso size have been tested as anticancer drug carriers, such as, for example, faujasite, beta-zeolites and pentasil, carrying 5-fluorouracil, ibuprofen, aspirin, diclofenac sodium, indomethacin, levofloxacin and doxorubicin hydrochloride. Nano-zeolite formulations are, however, still in an early research stage due to possible cytotoxicity arising from the alumina component and crystal shape. Alumina-containing nano-zeolites include, for example, the ZSM-5, LTL and LTA, all of which have surface acidic sites. These properties, it has been suggested, induce the observed necrosis of cells. In addition, surface charge plays an important role in nanoparticle toxicity: positive-charged surfaces are more toxic than negative-charged due to the strong affinity of cationic surfaces to negatively charged cell membranes [62]. As an example of nanosized zeolite carriers, 5-fluorouracil encapsulated in magnetite–zeolite (MZNC) nanocomposite particles was tested in vitro. These particles were cytotoxic to gastric tumor cells and caused up to 60% cell death. In addition, the transport and toxicity of the 5-fluorouracile to tumor cells was enhanced in comparison with 5-fluoruracile alone. The drug release from these magnetite–zeolite nanocomposites was also facilitated in the acidic pH [63]. Furthermore, the anticancer drug levofloxacin was prepared with a nanocomposite of zeolite-A and chitosan. The preparation showed itself to be an effective anti-inflammatory therapeutic approach that diminished cytokine production (IL-6 and IL-8) in the tested human bronchial epithelia cells. The safety of the zeolite-A/chitosan drug carrier preparation was corroborated by its low cytotoxicity [64]. Finally, immunotherapy might also benefit from nano-zeolite applications as carriers of monoclonal antibodies. Nano-sized zeolitic imidazolate frameworks (ZIFs) with encapsulated biopharmaceuticals have accordingly been studied in the last decade to target the death ligand-1 (anti-PD-L1) antibody delivery using a newly prepared ZIF-8 with PEG protection [65], while other ZIF-based nanocomposites that show good results in vitro on tumor cells and/or in vivo on mice, where they allowed a slow and continuous release of nivolumab and activated T cells/tumor-specific targeted delivery [66], enhanced the immune response rate of the therapeutic PD-L1 and CTLA-4 directed antibody [67], induced antigen-specific humoral and cytotoxic T lymphocyte responses by use of a newly prepared cancer vaccine based on an aluminum-integrated nanoscale metal–organic framework with antigen ovalbumin (OVA) packaged in a ZIF–8 nanomaterial [68], and proving efficient in photothermal therapy for cancer [69].

#### 2.1.4. Magnetic Nanoparticles

Magnetic NPs are studied for cancer therapy applications due to their potential in targeted drug delivery [70]. These particles are attractive as they produce more heat under microwave irradiation, which helps to release the loaded drug more easily [70]. Furthermore, magnetic drug delivery offers a long-circulating and furtive system which would not ultimately be eliminated by the phagocyte system. This feature can be acquired by a combination of peptides and polymers in the outer shell of the particle [71]. Generally, magnetic-based technologies allow for guided in vivo and in vitro drug delivery by functionalized nanoparticle surfaces with use of an external magnetic field. Superparamagnetic nanoparticles are the preferable choice for this kind of applications, as they have low aggregation properties and high magnetization energy activated by the external magnetic field [72]. Cautious usage is, however, advised as recently several clinical trials on superparamagnetic iron oxide nanoparticles were stopped due to their toxicity in vivo [73]. Some studies with iron oxide nanoparticles include a study of chemotherapeutic delivery to the lungs using an applied magnetic field [74], and delivery of starch-coated iron oxide NPs bound to the chemotherapeutic agent mitoxantrone directed towards the tumor site in an experimental rabbit model of squamous cell carcinoma [75]. Magnetic NPs have also been tested for targeted delivery of doxorubicin in an iron oxide NP shell with PEG and luteinizing-hormone-releasing hormone peptide. This specific preparation heated the cancer cells on site upon exposure to an alternating magnetic field [76]. Similarly, magnetic NPs, loaded with doxorubicin, functionalized with polymers and targeting the integrin β4 antibody, were tested for drug-release properties in vitro and for vehicle cytotoxicity on different tumor cells, where they showed no inherent cytotoxicity properties [77]. In addition, combined delivery of chemotherapy by magnetic NPs was achieved by the use of a single-component-polymer poly(vinyl alcohol) (PVA) shell stabilized by iron oxide NPs with encapsulated doxorubicin and paclitaxel. This delivery system resulted in controlled drug release in vitro and in vivo by the use of a high-frequency magnetic field trigger. Tumor cell death in vitro and tumor growth suppression in vivo when using the magneto-chemotherapy NPs loaded with these two chemotherapeutics was also confirmed, with little side effect [78]. Finally, an interesting application of NPs to cancer therapy might be envisaged in treatment options for brain tumors. As an attempt to contribute to the currently limited therapeutic options in this field, Kievit et al. prepared iron oxide core NPs coated with chitosan, PEG and PEI to deliver apurinic endonuclease siRNA to brain tumor cells. The choice of small-interfering RNA (siRNA) was based on the previously assessed role of apurinic endonuclease 1 enzyme in radiation resistance in cancer. Their results clearly showed decreased cell survival in clonogenic assays with pediatric brain tumor cells [79].

#### 2.1.5. Calcium Phosphate (CaP)-Based Mineral Systems

Due to a high similarity with bones, this material is highly biocompatible and has been explored in applications aimed at improving the issues of delivery and side effects with drugs used to treat cancer [80]. A number of in vivo studies using calcium phosphate-based NPs as the tested cancer therapeutic have been performed so far. The final endocytosis of such CaP particles into the tumor cells is usually dependent on the CaP NPs’ sizes, shapes and surface functionalization. For example, lipid–folic-acid–EGFR-specific single-chain fragment antibody functionalized CaP NPs were prepared and successfully tested for siRNA delivery to breast tumor cells in vitro and in vivo [81]. CaP-doped hollow mesoporous copper sulfide particles were also tested in breast tumor cells as inducers of Ca2+ for the disruption of mitochondria, which led to apoptosis due to the upregulation of caspase-3 and cytochrome c, and the downregulation of Bcl-2 and ATP. These effects were enhanced in vivo by the application of photothermal therapy [82]. For an overview of CaP-based NP systems tested in a clinical environment, please see Table 2.

#### 2.1.6. Carbon Nanoparticles

Carbon NPs (Figure 1) are constructed by the use of the element carbon. These NPs have an interesting potential for usage in medicine due to their exceptional physical and chemical properties [83]. In particular, they have been shown to easily penetrate cell membranes, and they display a particular electron hybridization of the carbon atoms that allows for wide possibilities of functionalization or loading with a therapeutic compound [84]. The carbon NPs include different types of carbon compounds such as, for example, graphene, graphyne, carbon nanotubes and fullerenes, with each of these categories bearing specific properties. Some of them have also been successfully tested in cancer therapy applications [85]. Graphene is, for example, a widely researched material in the area of drug design and drug delivery, but its use in medical applications remains limited due to its hydrophobic nature. In contrast, graphene derivatives, such as, for example, graphene oxide, show better properties for in vivo applications [83]. In particular, the epoxy and hydroxyl groups on the basal graphene plane, and the presence of chemical groups such as, for example, carbonyl, carboxyl or phenol structures, towards the material edges, allow for functionalization of the material for biomedical applications [86]. For example, co-delivery of doxorubicin and antimir-21 by graphene-oxide at a low doxorubicin dose had a strong antiproliferative effect on MDA-MB-231 breast cancer cells in vitro [87]. Moreover, high cytotoxicity to the tumor cells of CAL27, MG63 and HepG2 was observed for PEGylated graphene-oxide nanoparticles loaded with the anticancer drugs oridonin and methotrexate [88]. Graphene-oxide also showed a potential for targeting specifically cancer stem cells on an in vitro panel of breast, ovarian, prostate, lung, pancreatic cancer and glioblastoma tumor cell lines [89]. In recent years, graphene was also tested in animal models for the controlled delivery of anticancer drugs to mitochondria, but more research will be required, as these studies are at a very early stage [90].

Similarly, fullerenes were also tested for the treatment of cancer. These nanoparticles are formed of 60 carbon atoms, which form a hollow sphere or ellipse that can easily accommodate drugs in the hollow space [91]. Derivatization of fullerenes with polar groups has also been studied, especially due to their hydrophobic nature. Besides hydrophobicity, these NPs also have a high photoactivity and reactivity due to their property of accepting and releasing electrons (antioxidative effect and/or ROS production in cells), which can be exploited in anticancer therapy [91]. For example, fullerenes may be used in photodynamic cancer therapy [92], and they were used as tumor targeting/therapeutic agents in five mouse models, including H22 hepatocarcinoma, human lung giant cell carcinoma PD, human colon cancer HCT-8, human gastric cancer MGC803 and human OS732 osteosarcoma, where the accumulation of fullerene-NPs in the tumors was enhanced, which is highly relevant for the photodynamic therapy of tumors [93]. In addition, fullerenes were tested as cisplatin-loading NPs for the reversing of tumor resistance to cisplatin in cisplatin-resistant human prostate cancer (CP-r) cells [94]. Finally, carbon nanotubes (CNTs), cylindrical tubes known also as rolls of carbon sheets, have been tested as immune-response-triggering NPs for suppressing tumor growth [95] and as a promising 5-fluorouracile delivery system on MCF-7 breast cancer cells in vitro [96]. Still, these CNTs have a number of limitations in medical application as they are hydrophobic, have a low biodegradability, may easily interact with the biomolecules which underlie their toxicity, and there are even possible effects on the genome [84].

### 2.2. Organic Nanoparticles

Even though inorganic nanoparticles have gained substantial attention in the area of cancer therapy, their toxicity and the limitations for safe in vivo usage have also prompted a high level of interest in the study of organic, biocompatible nanoparticles for different cancer treatment applications. Such organic nanoparticles are developed in such a way as to be nontoxic for cells and biodegradable and to not induce cellular or tissue damage. Among such molecules are polymers (Figure 2), nanogels, nanofibers, micelles, liposomes and extracellular vesicles [97].

#### 2.2.1. Polymer, Nanogel and Nanofiber Nanoparticles

Polymeric nanoparticles are highly stable, colloidal, biocompatible and biodegradable nanomaterials that encapsulate hydrophobic chemotherapeutics in their matrices. Encapsulation of the drug in the matrix improves its bioavailability in the cells [98]. These NPs have the intrinsic property of sustained and controlled drug release. Furthermore, polymeric NPs provide many advantages, such as sustained drug release, prevention of drug detoxification and metabolism, avoidance of systemic clearance, longer circulation time and enhancement of intracellular uptake [99]. There are many studies on how to modulate ATP-binding cassette (ABC) efflux transporters and improve the intracellular accumulation of chemotherapeutics by loading them in polymeric nanoparticles. Moreover, Park et al. [100] showed that PLGA NPs may be used successfully to encapsulate Adriamycin. In addition, a repurposed drug for anticancer applications, namely, verteporfin, may also be delivered to tumor cells by the use of PLGA NPs, as shown by Shah et al. on an in vivo breast cancer mice model [101]. Chitosan NPs were also widely tested in recent years in anticancer applications, as they allow for a controlled release of drugs upon its degradation in vivo with no toxicity effects, good cellular uptake properties as a result of their amphiphilic chain that interacts with cell membranes, and a long circulation time and pH selectivity for cancer cells [102]. For example, hyaluronic acid–chitosan nanoparticles were used for a co-delivery of doxorubicin and miR-34a, showing a synergistic antitumor effect in triple-negative breast cancer models both in vitro and in vivo [103]. Moreover, chitosan NPs were successfully tested as delivery agents of docetaxel in lung cancer [104] and celecoxib in colon cancer [105]. Polymeric nanoparticles are often coated with polysorbates that help polymeric nanomaterials interact with BBB and endothelial cell membranes and to facilitate endocytosis [106]. Recently, fluorescent polymeric nanoparticles have been used as a theragnostic tool. Theragnostics combines diagnosis and treatment at the same time [107]. Polymeric nanoparticles sensitive to ultrasound have become an effective tool for cancer diagnosis and treatment. Ultrasound helps to improve drug delivery efficiency, leading to reduced side effects through an improved permeability to overcome barriers in cancer therapy [108]. It can also be used as a preset trigger, to eventually break up the nanoparticles and release the drugs under control [109]. Similarly, specific nano-formulations, known as organic, biocompatible nanogels, have been widely tested in anticancer applications. Nanogels are non-toxic, biocompatible porous structures with a large surface area and drug loading capacity particularly suitable for topical applications in clinics. For example, there are chitosan-based nanogels for local delivery with increased mucosal exposure to doxorubicin in colorectal cancer [110]. Recently, there have been nanogel formulations, such as, for example, a self-assembled polysaccharide nanogel of cholesteryl-group-modified pullulan for cancer vaccine delivery and enhancement of the immune response against tumor cells [111]. Finally, nanofibers are extremely interesting nanostructures for precise and controlled cancer cell targeting and treatment. These solid fibers ranging from a few nanometers to 1000 nm in diameter can be engineered in such a way as to obtain different drug release kinetics. Natural nanofibers include bacterial cellulose and silk fibroin nanofibers [112]. Nanofibers are particularly interesting as devices for implantation and localized drug delivery in postsurgical sites after tumor removal, to eliminate remaining cancer cells or to avoid cancer relapse, or in the treatment of tumors that are not easily resected surgically, i.e., pancreatic cancer. Various polymers are used for nanofiber production, including PVA, polyethylene, poly(lactic acid) (PLA) and PLGA [113]. Some nanofibers have already been prepared for the delivery of temozolomide and paclitaxel against glioblastoma cancer cells. The nanofibers encapsulated more than 80% of the chemotherapeutic drugs, which makes them attractive for use as drug carriers [114]. Indeed, nanofibers are particularly suitable for brain tumor treatment such as, for example, PLGA electro-spun fibers for sustained paclitaxel release that have shown promising results in mice with glioma [115].

#### 2.2.2. Liposomes

The first clinically approved nano-systems for anticancer drug delivery [116], liposomal nano-formulations, are spherical vesicles that include amphiphilic phospholipids and cholesterol associated with an aqueous lumen [117]. This is the reason why liposomes can encapsulate both hydrophobic and hydrophilic chemotherapeutics within their core. Still, liposomes may be unstable in vivo as some types of bilayers undergo oxidation or hydrolysis, but these issues may be overcome by careful choice of the material used for liposome production, by a directed and controlled production process and by delivery of the formulation in the lyophilized form [118]. Liposomes used in anticancer therapy may be produced to carry synergistic drugs, to exert a pH-dependent drug release, as light-, ultrasound- or magnetic-sensitive liposomes or liposome-in-gel formulations. All these nanoparticles have shown improved delivery, increased targeting properties towards tumor cells, improved safety profile and increased therapeutic efficacy for the drugs encapsulated within the liposome nanoparticles [119]. Application of liposomal drug delivery systems was also successfully tested for overcoming multidrug resistance. For example, Tang et al. co-encapsulated doxorubicin and verapamil in a liposomal nano-formulation which overcame P-gp-mediated MDR in human breast cancer cells with reduced toxicity in non-target organs [120]. Several liposomal nano-formulations of chemotherapeutics are under clinical study and some of them are already approved by the FDA for cancer treatment (Table 3). One of the approved drugs is nanoliposome Vyxeous, which has been used to co-deliver cytarabine and daunorubicin for the treatment of acute myeloid leukemia (AML) [121].

#### 2.2.3. Micelles

Micelles are self-assembled hydrophilic and hydrophobic blocks in an aqueous environment with a hydrophobic core. The hydrophobic core is able to entrap hydrophobic chemotherapeutics, while the hydrophilic outer shells enable prolonged circulation time and accumulation in the tumor tissue via the enhanced EPR mechanism [122]. Polymers, such as poly (aspartic acid), poly (caprolactone) (PCL), PLGA and PEG, are used to form the micelles [117]. There are several experimental and clinical studies that have been evaluating polymeric micelles loaded with chemotherapeutics for their anticancer effect. For instance, Lv et al. demonstrated the use of polymeric micelles (PEG2k-PLA5k) to co-deliver doxorubicin with curcumin to reverse MDR via dual-drug-based nano-micelles in drug-resistant MCF-7/ADR cells and in a xenograft model [123]. Many different polymeric nano-micelles have achieved great success at different clinical stages. For example, one of them is Genexol-PM. It consists of nano-micelles loaded with paclitaxel and it has been approved by the FDA for the treatment of breast cancer patients. Studies in a preclinical in vivo phase displayed a threefold rise in the maximum tolerated dose of paclitaxel and increased antitumor activity in comparison to the drug in free form [122]. Finally, polyplex micelles may also be of interest in cancer treatment for the delivery of hydrophobic anticancer drugs, otherwise applied with surfactants or even organic solvents. These particular types of micelles are PEG-shielded gene delivery systems formulated upon poly-ionic complexation-induced self-assembly between PEG-polycation block copolymers and plasmid (p)DNA. Polyplex micelles have been, accordingly, tested as gene vectors as an alternative to viral vectors and showed positive therapeutic outcomes in certain cancers’ treatment observed so far [124,125]. For example, cyclic RGD (Arg-Gly-Asp) peptide ligand decorated polyplex micelle loading pDNA encoding the human soluble form of vascular endothelial growth factor receptor-1 (or soluble fms-like tyrosine kinase-1) mediated αvβ3 and αvβ5 integrin-mediated uptake and showed anti-tumor activity against subcutaneously xenograftedBxPC3 human pancreatic adenocarcinoma in mice [126].

#### 2.2.4. Extracellular Vesicles

Extracellular vesicles (EV) are bilayer phospholipid vesicles of different sizes, origins and contents, which are continuously secreted by different cells [127]. Based on their origin, EVs are classified into three major groups: exosomes, micro-vesicles and apoptotic bodies. EVs contain proteins, RNA and DNA and they are involved in long-distance communication [128]. Due to the resemblance of exosome membranes’ lipids to the origin cells, exosome nanoparticles have the ability to evade immune surveillance and internalize with target cells, making them perfect for use as drug carriers [127]. Scientists have begun to use exosomes as nanoparticle platforms to deliver nucleic acids, small molecules and proteins [129]. In their research, Hadla et al. loaded exosomes with doxorubicin and used them to treat human breast cancer cells. The results showed an decreased cytotoxicity compared to the free drug, and accumulation of the drug in the heart was avoided [129]. Unlike synthetic nanoparticles, exosomes have inherent biocompatibility, greater chemical stability and the ability to control intercellular communications. However, exosome nanoparticles have their own disadvantages and limitations, such as the lack of unique criteria for exosomal isolation and purification, an unclear mechanism of exosomes in cancer treatment, heterogeneity and difficulties in preservation [130].

## 3. Intracellular Transport of Nanoparticles

### 3.1. Passive Transport of Nanoparticles into the Tumor

NPs have shown benefits due to their ability to capture the small drug molecules inside their structure and therefore prolong the time that they remain inside the tumor [131]. This previously described infiltration/penetration of the NPs into the tumor tissue is called passive entrance. This type of intratumor transport relies on the NPs’ size and circulation time, as well as the permeability and vascularity of tumor tissue [132]. Some of the examples of widely used nanocarriers which intrude into the tumor via passive intracellular transport are Doxil, the first FDA-approved nano-drug, and Caelyx, a liposomal nano-drug [133,134]. The passive intratumor transport of NPs is presented in Figure 3.

As previously elaborated, the EPR effect underlying the passive intratumor targeting properties of NPs is highly dependent on the intrinsic tumor biology, which includes the stage of angiogenesis and lymph-angiogenesis, perivascular tumor growth and intratumor pressure [137]. Another problem that occurs is that, due to irregular vessel growth and arrangement, even tumor cells grow irregularly, and some are more developed and some are not. NPs do not reach those that are poorly developed, because of the lack of nutrient and oxygen supply [138]. Moreover, a phenomenon that is observed in the central part of the tumor is that the blood vessels have lower permeability in comparison with those around the center of the tumor due to the high interstitial pressure. This high interstitial pressure is present in many types of solid tumors [138]. This causes the inhibition of drug delivery in the center of the tumor and pushes the blood vessels towards the periphery of the tumor [132]. A way to bypass these problems is to apply EPR mechanism enhancers such as bradykinin, nitric oxide, peroxy-nitrite, prostaglandins, vascular permeability factor (VPF) and/or vascular endothelial growth factor (VEGF), and other cytokines or macromolecules [139]. These enhancers induce hypertension and/or vessel normalization, which could possibly enhance tumor overflow. Besides EPR enhancers, there are other approaches such as radiation, ultrasound, hyperthermia or photoimmunotherapy, which can also increase NPs’ infusion, as they allow for tumor-selective combination therapy by guided physical approaches such as, for example, multimodal-imaging-guided tumor inhibition. All of these methods ought to be applied carefully, to ensure safety and efficiency [131]. Additionally, surface ligands and charge are also important in the circulation time. For instance, NPs that are too hydrophobic or charged are quickly opsonized by the mucopolysaccharidoses, and because of this NPs should have an anionic or neutral charge with a hydrophilic property [140].

### 3.2. Active Cell Targeting by Nanoparticles

Active intracellular transport ought to be a more precise means of drug delivery in the targeted tissue. It is programed to target specific cells and therefore the NPs’ accumulation, retention and delivery rates are enhanced [141]. Since active transport of NPs does not depend on EPR, this allows targeting of hematological malignancies, metastatic tumors and many other diseases that cannot be affected via passive transport [142]. This kind of strategy increases the drug penetration into the targeted cells. In the 1980s, antibodies were the first ligands examined to bind to NP surfaces; however, in more recent years, a wide range of ligands, such as peptides, nucleic acids, sugars and small molecules, have been used. Physical ligand–target interaction triggers the unfolding of the membrane and internalization via endocytosis [143]. In Figure 4, one of the mechanisms of the interaction of NPs carrying drugs into the tumor cell is presented. After monoclonal antibody recognition with a binding site on the tumor cell surface, the endocytosis process is initiated and the NPs carrying the drug enter the cell. When referring to active transport NPs to the target cells, it should be taken into account that one needs to find a perfect balance of the dissociation constant. Several studies have shown that ligands that are placed tightly near each other have a greater binding effect on the targeted tissue and that their mutual connection is much stronger [144,145]. Binding affinity is defined as the strength of a molecule to bind with a targeted counterpart molecule. An example of increased binding affinity is that of NPs that have more folates on their surface; however, this has a limit. Binding affinities can also be decreased due to a very high concentration of ligands on the NP’s surface. The reason for this is that there are many steric binding interferences that ultimately prevent the binding of a ligand to the antigen. This is an issue that should be taken into account when generating novel NPs for drug delivery and active intracellular targeting [145]. The bonding through which the binding affinity is based includes various chemical forces, such as the hydrophobic effect, ionic bonds and hydrogen bonds, as well as Van der Waals interactions. Among these, the Van der Waals forces are the weakest among the chemical interactions, compared to hydrogen bonds and ionic bonds. The ionic bonds have a stronger ability to bind two molecules through the mechanisms of opposite charge attraction and interaction [141]. The targeting agents are the main components of NPs that are made for active cell targeting. Each ligand on an NP’s surface has its own advantages and disadvantages. The main advantages of these ligands are the positive effects on NPs’ transmembrane penetration, accumulation in the tissues where the EPR effect is not present, a more effective therapeutics delivery and imaging. The ligands used for the functionalization of nanoparticles include popular and well-studied synthetic polymers, small molecules, biomacromolecules including organic–inorganic hybrid NPs, silica and gold [146]. As an example, magnetic inorganic nanoparticles might be functionalized for cell targeting by thiols, phosphonates and carboxylic acid chemical groups on the surface [147]. A great value of active cell targeting by NPs has been proven, especially in cancers that do not manifest the EPR effect, such as some hematological malignancies, small tumors that metastasize and circulating tumor cells [148,149,150,151]. For instance, antibody-modified iron oxide NPs targeted to the HER2 receptor (human epidermal growth factor receptor 2) were capable of binding to breast cancer metastases in the liver, lungs, brain and bone marrow in mouse models [152]. Moreover, an even better effect of the NPs is accomplished when such functionalized NPs are loaded with chemotherapeutic agents [145]. Another example of active targeting NPs’ benefit is leukemia. In acute myeloid leukemia (AML), epidermal growth factor receptor (EFGR) is overly expressed on malignant cells, which can be easily targeted by an anti-EGFR antibody. A study by Durfee et al. [142] has shown success in binding actively targeted NPs to the EGFR-expressed leukemia cells in both in vivo and ex vivo models. By this action, anti-EGFR-antibody-targeted NPs were proven to induce leukemia cell death even more successfully [153]. Transferrin-modified PEG-phosphatidyl-ethanolamine NPs that were specifically designed to target ovarian carcinoma cells expressing transferrin showed similar success [154]. Furthermore, a surface of the tumor endothelium cells expresses glycoprotein known as vascular cell adhesion molecule-1 (VCAM-1), which is actively involved in the process of angiogenesis. Research has shown that NPs that target VCAM-1 in a breast cancer model could potentially be beneficial [155]. Additionally, cancer cells overexpress FR-α (alpha isoform of folate receptor), while its variation FR-β (beta isoform of folate receptor) is overexpressed in liquid cancer cells [156].

Despite the substantial research with actively targeted NPs, no such NP system is being used in clinics. A significant part of the active NPs targeting tumor cells have shown great success in phase 1 and phase 2 clinical trials, but none has passed the crucial phase 3 of clinical trials [141].

## 4. Current Usage of Nanoparticles

Currently, in clinical settings, the only approved NPs are those that enter the cells via passive intracellular transport. The previously mentioned NPs that enter through active intracellular transport have not yet passed the final clinical phase. Therefore, when it comes to treatment, current usage of NPs in medicine is focused on cell imaging, in vivo imaging, gene delivery, drug delivery, cancer treatment and regenerative medicine. A schematic overview is shown in Figure 5 [127]. When it comes to cancer treatment, NPs show great potential benefit due to their specific properties. For example, photodynamic (PDT) and photothermal therapy (PTT) are two treatment methods that are related to optical interference. In PDT, photosensitizer is accumulated in cancerous sites when irradiated with a certain wavelength of light. After the irradiation, singlet oxygen and other cytotoxic reactive oxygen materials are produced, which ultimately induce cancer cell apoptosis [157]. On the other hand, PTT uses materials that have high photothermal conversion efficiency to raise the temperature of the targeted cancer areas, which leads to cell apoptosis. One of the main obstacles is the toxicity of nanomaterials. They are extremely small in size and, therefore, physiological barriers can be easily penetrated, which can cause potential health hazards. From the potential toxicity of NPs caused by free radicals, cell membranes, organelles and DNA materials can be damaged [158]. Some of the nanoparticles’ toxicity mechanisms include the physical disruption of cell membranes, structural changes in the cytoskeleton, oxidative damage to DNA and impaired transcription, damage to mitochondria, disruption of lysosome trafficking, ROS production and pro-inflammatory effects [159]. Another problem is that the research which is done on NPs cannot mimic the real human organism and every physiological occurrence. That is the reason why most of the tested NPs cannot go further than the final, clinical stage of research [160]. For this reason, all the qualitative and quantitative physical and chemical properties of NPs are systematically studied for each type of produced nanoparticle, in order to understand the possible effects on biological systems.

## 5. Nanotechnology-Mediated Cancer Treatment

Due to the unique properties and features of nanoparticles, such as size, composition (lipids, dextran, lactic acid, phospholipids, chitosan) and chemicals (silica, carbon, metals and different polymers), their interactions are different from those of non-biological components. Nanoparticles must be non-toxic, chemically stable and biocompatible so they can be efficient drug delivery agents. Given that the delivery of nanotherapeutic platforms depends primarily on the passive targeting of tumors, it is necessary to either bridge or strengthen this mechanism. Further, to change the timing or site of drug release, other methods, such as radiotherapy, ultrasound, changes in pH and temperature, are used and combined with standard anticancer drugs [161]. Table 4 contains a list of nano-formulations that have been used or are currently part of oncology practice for the treatment of patients with malignant diseases. Sriraman et al. [162], for example, found that a new PEGylated liposome carrying cobimetinib and ncl-240 displayed an enhanced synergistic cytotoxic effect of these compounds. Similarly, other authors have shown improved anti-cancer effects of floxuridine and irinotecan or CPX-1 liposome injection when entrapped in liposomes for the treatment of breast cancer and solid tumors, respectively [163,164].

Nanoparticle vaccines are also being designed to raise the T cell response in order to fight tumors, to stimulate dendritic cells and targets, and for the release of antigens for stimulation of the immune system against tumor cells [165]. Accordingly, nanoparticles are now tested in clinical trials as vehicles or drug combinations for immunotherapeutic and immunomodulatory agents in the form of vaccines, cytokines and adoptive cellular therapies [166,167,168]. In a melanoma study, liposome-based particles loaded with a model tumor antigen with a Toll-like receptor agonist showed a better immunological response against the tumor in comparison with a vaccination method. In addition, dendritic cell nano-therapy has been tested in melanoma, prostate cancer, hepatocellular carcinoma and renal cell cancer clinical trials with a still low efficiency outcome [169,170]. Moreover, almost 50–60% of all cancer patients receive some form of radiation therapy during their cancer treatment period. Radiation can be extremely toxic, not only to cancer cells but to normal, healthy cells, which dramatically limits its use [5]. Combined nanotechnology and radiotherapy effects rely on the interaction between X-rays and nanoparticles due to the inherent atomic-level properties of the materials. The first mechanism of the interaction between X-rays and NPs includes the NPs with high atomic number Z that enhance the photoelectric and Compton effects, which leads to the emission of secondary electrons that add to tumor cell destruction, in order to increase the efficacy of conventional radiation therapy [171]. The second mechanism of action is based on drug release from the NPs on the tumor site. One example of where NPs for enhanced radio-sensitization can be employed in a clinical practice is for the treatment of glioblastoma, where high-Z metal NPs have been already successfully tested in animal models [171]. Recently, gene therapy for cancer has also gained interest. A large number of studies have researched the potential of nanomaterial-based drug delivery for DNA- and RNA-based genetic therapeutics delivery, including genes, gene segments, oligonucleotides, siRNAs and microRNAs (miRNAs), both ex vivo and in vivo. The use of NPs for the delivery of gene-based therapeutics allows for good loading capacity, efficient targeting of the tumor cells and avoidance of the host immune system [172]. In cancer therapy, this group of therapeutics is prepared in a nano-formulation, and it has been tested for enhancement of the host immune system towards cancer, the silencing of oncogenes and induction of tumor-suppressors, the so-called suicide-gene therapy, and for the modulation of miRNA processes and transcription. Some of these strategies are used for targeting well-studied cancer-associated proteins that are ‘undruggable’, i.e., myc, p53 and RAS. One of the strategies in targeting these cancer-associated proteins is the use of silencing molecules, siRNAs or short-hairpin RNAs (shRNAs). For example, in a study by Conde et al., c-Myc was targeted with glycoNPs-siRNA in vivo on a lung cancer mice model which caused an approximate reduction in the tumor size of up to 80% [173]. Gold NPs with PEG or PEI with transferrin- or folate-receptor-targeting ligands were also successfully tested for the delivery of siRNA into prostate tumor cells in vitro as part of a strategy to enhance the overexpression of the prostate-specific membrane antigen [174]. miRNA molecules have also been successfully delivered to tumor cells by the use of different NPs. A particularly important achievement has been made in testing novel possibilities for pancreatic cancer treatment. For example, cationic β-cyclodextrin-PEI NPs delivering miR-34a for targeted activity on E2F Transcription Factor 3 (E2F3), B-cell lymphoma 2 (Bcl-2) and c-myc genes, proved a valuable tool for efficient tumor treatment in a PANC-1 xenograft model [175]. Similarly, a combined therapy approach using microRNA-21 antisense oligonucleotides and gemcitabine with targeted co-delivery by the use of a PEG/PEI- iron oxide NP on pancreatic cancer cell metastasis and growth had promising results [176]. This approach in targeting tumor-associated proteins and processes has also found its way into clinical trials, and a study of combined gene construct SGT-53 plus gemcitabine/nab-paclitaxel for metastatic pancreatic cancer is ongoing until the end of 2023 [177]. The silencing approach might be used in applications for overcoming the mechanisms of multidrug resistance during tumor therapy. For example, Nieth et al. [178] developed an MDR-resistant human gastric carcinoma cell line for treatment with a liposomal formulation of the anti-MDR1 siRNAs. They observed substantial inhibition of the MDR1 expression up to 91% at the mRNA and protein levels, which caused restoration of the tumor cells’ sensitivity to daunorubicin. Moreover, siRNA was delivered to vemurafenib-resistant BRAF mutated melanoma by use of polymetformin NPs for the direct suppression of drug resistance. Gene editing methods, such as, for example, CRISPR/Cas9 (clustered regularly interspaced short palindromic repeats and CRISPR-associated protein 9), have been successfully tested combined with liposomal NPs for targeted tumor cell delivery [179]. In particular, a combined NP delivery of chemotherapy along with wild-type p53 or a CRISPR/Cas9 gene-editing system may be regarded as an interesting anticancer approach [122,123]. Similarly, CpG oligonucleotides have been successfully prepared in DNA nanostructures to increase their cell uptake efficiency. The use of CpG oligonucleotides in the treatment of tumors induces mammalian innate immune systems, which is particularly true for synthetic CpGs that stimulate the activity of the Toll-like-receptor 9 (TLR9) and the secretion of pro-inflammatory cytokines [180]. Some application issues, such as, for example, enzyme digestion and metabolization of the NPs, have also been studied and partially solved [181,182]. For example, improvements were achieved with the development of stable organic and lipid-based nanoparticles, and inorganic nanoparticles, such as QDs, AuNPs and silica-based nanoparticles [161,183]. Finally, the combination of viral particles with nanoparticles provides additional possibilities regarding better biocompatibility and the use of host resources for delivering chemotherapy to tumors, especially if the system is also applying PEGylation to enhance viral survival from the immune system [184]. Virus-like particles (VLPs), such as the cowpea mosaic virus, have recently become a focus of interest because of their characteristic that they do not contain nucleic acid material and therefore can be safely introduced into a body [185]. At the same time, it was shown that VLPs stimulate the strengthening or induction of the body’s immune response due to increased releases of large amounts of neutrophils and the consequent appearance of recruit cytokines and T lymphocytes, which are subsequently involved in the immune attack on tumor cells [184].

## 6. New Developments in Nanomedicine: Nanobubbles

As has already been discussed in great detail in this review, the application of nanomedicine in cancer detection, treatment and prevention has great potential, and yet many clinical trials have shown only weak to moderate effects [186]. One important new development in the area is, accordingly, nanobubbles. These NPs were first applied in diagnostics but their use was expanded to research in the areas of cancer prevention, direct treatment, enhanced delivery of siRNA, miRNA and CRISPR/cas9, and precision medicine with the “theragnostics” explained earlier. Enhanced drug delivery and EPR have had some success with nanobubbles, too [187,188]. We will briefly discuss some of these new developments.

### 6.1. Fundamental Studies and Generation of Nanobubbles

Gas/liquid interfaces are important in many areas of technology and medicine, including diagnostics. Macrobubbles with particle sizes over 50 microns are quite common. Microbubbles (15–50 microns) are a more recent development. In the 1990s, however, nanobubbles (1–1000 nm) became a hot topic of research and shortly thereafter were applied in many fields [187]. In short, classical physics, including the Laplace equation and Epstein–Plesset theory, states that nanobubbles, due to their small size, should not exist, and gas introduced into liquid will either dissolve or create microbubbles and macrobubbles that rise to the top of the vessel [187,189]. Nanobubbles adsorb hydroxyl ions from solutions and, in part due to that surface charge, are stable for weeks or even months. However, in medicine, nanobubbles of hydrogen are quite popular. Hydrogen molecules are very small and so are nanobubbles of hydrogen. They can diffuse through plastic containers and are best used after they are generated. Alternatively, a patient can inhale a gas containing some hydrogen [188]. Lung surfactant lowers surface tension and creates a perfect opportunity to create nanobubbles. Later, nanobubbles enter the bloodstream where they encounter albumin and other proteins that keep the surface tension of water low. One method of nanobubble generation is pressure cavitation through small orifices [189]. Our heart is just such a pump, and tight junctions inside the cells are ideal small cavities. For nanomedicine purposes, the most popular means of nanobubble generation are electrolysis with a semipermeable membrane keeping protons and hydroxyl ions separated, sparging under pressure through ceramic porous nanofiltration membranes, high-energy mixing and ultrasound with high-energy sonication tips or electromagnetic radiation such as RF [187,189,190]. Many different gases can be used, such as hydrogen, oxygen, carbon dioxide, SF6, etc. Nanobubbles can entrap many drugs inside the hydrophobic core, as well as siRNA, CRISPR/cas9, photosensitizers, acoustic sensitizers, etc. [187]. On the outside, nanobubbles can be coated with lipids, surfactants, proteins, nanoparticles, polymers, antibodies, etc. [191], and if oxygen is used it can be employed to target tumor hypoxia and even produce ROS directly inside the tumor tissues.

### 6.2. Application of “Naked” Nanobubbles of Hydrogen in Cancer Treatment and Side Effects Reduction

The application of hydrogen nanobubbles in cancer treatment has recently been reviewed in [188] and will only be briefly discussed here. Electrolyzed water machines are quite popular, and millions have been sold worldwide. However, they produce no more than 0.3 mg/L and 106/mL hydrogen nanobubbles.

Hydrogen sparging through ceramic membranes or, even better, hydrogen containing gas inhalation did produce the best results, as indicated in [188]. However, the results are still modest: life extension of up to 9 months has been achieved, but this happens rarely. The most impressive effects were observed with the treatment of non-small cell lung cancers (NSCLCs). Recent reports that hydrogen inhalation enhanced the performance of nivolumab—antibody against programmed cell death protein 1 (PD-1) at exhausted CD8+ T cells is most impressive [192]. This happened, like many other activities of hydrogen in molecular medicine, through the inactivation of IGF-1 (insulin-like growth factor 1)—protein kinase Akt—mTOR signal transduction cascade in the mitochondria of exhausted cells and the activation of protein kinase AMPK.

### 6.3. Mechanism of Anticancer Action of “Naked” Hydrogen Nanobubbles

Molecular fully dissolved hydrogen is biologically mostly inactive. The nanobubble action in many different areas was long a puzzle and is still an intensive area of research. It was proposed in 1998 that nanobubbles react with water, agglomerate and burst periodically, and produce hydroxyl radicals and atomic hydrogen through quantum vacuum radiation effects [190]. Cations present in cells, such as ferric, and copper or zinc, can catalyze such free radicals’ production [192,193,194]. A small amount of hydroxyl radical then stimulates cells’ antioxidants defense activation via Nrf2 (Nuclear factor erythroid 2-related factor 2) transcription factor activation. Atomic hydrogen as a potent antioxidant participates in the inhibition of the IGF-1—protein kinase Akt- mTOR signal transduction cascade and activation of AMPK protein kinase. The activity of protein kinases ERK1, MAPK and p38 also seems to be modified. Atomic hydrogen can be stabilized, encaged in many molecules and particles such as methyl cobalamin or microporous zeolites [195]. Hydrogen nanobubbles are so small (smaller than 100 nm, often below 20 nm) that they can penetrate almost every tissue including the BBB. The problems with nanoparticle delivery discussed in Chapter 3 can be dealt with by using such very small hydrogen nanobubbles.

### 6.4. Armored Oxygen Nanobubbles [191]

As explained in Section 3, due to the fast growth of tumors, an imbalance in oxygen supply and demand often occurs, resulting in hypoxia or anoxia, particularly inside a solid tumor body. Oxygen nanobubbles are a perfect tool to fight such hypoxia. Naked oxygen nanobubbles have been used successfully for such a purpose [196]. Animals drinking hydrogen-nanobubble-enriched water have shown a shrinkage in tumor cells. In the in vitro assays described in [196], hypoxia-induced factor 1 (HIF1) was inactivated, along with increased VEGF and N-α-acetyltransferase ARD1 homolog A protein (ARD1A). Armor shells can be lipids, surfactant micelles, emulsions, proteins, or polymers, and even nanoparticles as described previously.

Over 100 manuscripts have also appeared on the topic of applications of nanobubbles in genetical engineering. For example, biosynthetic nanobubble-mediated CRSPR/cas9 gene editing of Cadherin 2-mediated inhibition of breast cancer metastasis has been described [197]. Many manuscripts describe nanobubbles with ultrasound-enhanced siRNA delivery [198]. In [198], nanobubbles loaded with siRNA were used to antagonize drug resistance for NSCLC. Chitosan nanobubbles have been used to deliver an miRNA antagonist to miRNA-17 to treat B-cells lymphoma. The application of microbubbles and nanobubbles with ultrasound for systemic gene delivery was reviewed in [199]. Even the combination of mir-424 RNA with PD-L1 antibody has been described to inhibit the growth of hepatocellular carcinoma in mice with enhanced immunotherapy [200]. Mechanistic insights into therapeutic gene delivery through microbubbles and nanobubbles assisted with powerful ultrasound were reviewed in [201,202]. A comparison of the therapeutic effects of microbubbles, nanobubbles and nanobubbles assisted with pointed ultrasound irradiation has been performed [203]. Gliomas are among the most invasive, resistant and infiltrative tumors. The application of siRNA or gene therapy has been attempted but failed. Cationic surfactants and lipids are very toxic, viral vectors are immunogenic and the siRNA is very sensitive. Microbubble-assisted delivery yielded very poor results. However, nanobubble-assisted delivery with pointed ultrasound showed excellent results both in vitro and in vivo in a mice model [203]. The EPR was excellent. Nanobubbles combined with ultrasound penetrated the leaky tumor blood vessels easily. The cell–cell distance in the tumor blood vessels was between 380 and 780 nm, versus only 7 nm in normal tissue. The nanobubbles were coated with avidin–biotin with siRNA. Excellent results were achieved in both the reduction of the tumor growth rate and in the long-term survival rate of the mice [203]. To conclude, applications of nanobubbles with targeted ultrasound irradiation is a very promising diagnostic and therapeutic (theragnostic) tool [202].

## 7. Future Perspectives

Although nanotechnology has great potential, there is still no NP-based cancer blockbuster on the market today comparable to Revlimid^®^, Opdivo^®^, Imbruvica^®^, Keytruda^®^, Texcentriq^®^, Perjeta^®^, etc. Characteristics of nanoparticles, such as their small size, high reactivity, and the unique tensile and magnetic properties of nanomaterials, have raised concerns about the implications for health and safety. As just one example, the issue of the toxicity of CNTs observed in animal studies is still not clarified in detail for in vivo systems. However, considering the very sophisticated technology that is used today in the development and production of NPs, as well as the increasingly rigorous means for controlling the potential toxicity of new materials, it is expected that every new nanotechnology product will be very safe for use. The characteristics that every new nanotechnology agent must have are reliability, reproducibility, being highly sensitive and specific with a stable structure, being easy to handle and, of course, cost effectiveness. However, we must note that our current approaches, whether they are advanced immunotherapy, gene therapy or an “integrative medicine” that includes nanobubbles, nanoparticles and nutraceuticals, are highly unacceptable [186]. This is in part due to the inadequate animal model systems and general R&D approaches being used [186]. Most of the tested and approved drugs and therapies at best extend patients’ lives from 1 to 18 months, or reduce some of the side-effects of chemotherapy, radiation or surgery. Synergistic, multimodal therapy that will include nanomedicine is badly needed both for therapy and diagnostics. The so-called theragnostics discussed earlier shows great promise. Precision medicine along with a personalized approach is very important. Most tumors are very different and often unique. Our ability to identify the small number of cancer cells that survive treatment is almost non-existent. The National Institutes of Health Cancer Genome Atlas has characterized over 20,000 primary cancers across 33 cancer types. Genomics data are accompanied by epigenomics, transcriptomics, proteomics and, more recently, by some single-cell spatially resolved multi-omics. The ability of cancer cells to mutate and learn how to fight any applied therapy is just short of miraculous. Thinking “out of the box” is the only way to improve our current therapeutics approach. We have barely started to address cancer stem cells and resistant metastatic tumors. A long road still lies ahead of researchers to develop a broad array of safe NP-based therapeutic systems for cancer treatment.

## Figures and Tables

**Figure 1 ijms-24-12827-f001:**
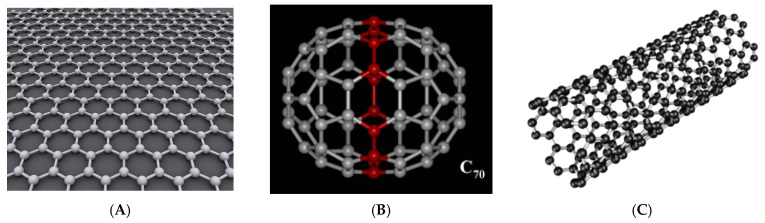
Depictions of carbon nanomaterials: graphene hexagonal grid (**A**), fullerene C70 (**B**) and carbon nanotube (**C**). Carbon nanomaterials are structures made up of well-arranged sp2 carbon atoms and a nanoscale diameter with one or more walls or sheets. Figures provided by Wikimedia Commons and available under the Creative Commons CC0 License and Creative Commons Attribution-Share Alike 3.0.

**Figure 2 ijms-24-12827-f002:**
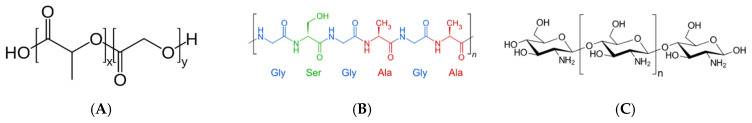
Examples of polymers used for preparing polymer-based nanoparticles. Poly(lactic-co-glycolic) acid PLGA is a biocompatible synthetic polymer, while chitosan and fibroin are natural materials. Skeletal formula of chitosan—a linear polysaccharide composed of randomly distributed β-(1 → 4)-linked d-glucosamine (deacetylated unit) and N-acetyl-d-glucosamine (acetylated unit). This structure shows completely deacetylated chitosan. Figures provided by Wikimedia Commons and available under the Creative Commons CC0 License and Creative Commons Attribution-Share Alike 3.0. (**A**) Skeletal formula of poly(lactic-co-glycolic acid) (PLGA). Created using ACD/ChemSketch 10.0 and Inkscape. PLGA. (**B**) Silk fibroin primary structure. (**C**) Chitosan.

**Figure 3 ijms-24-12827-f003:**
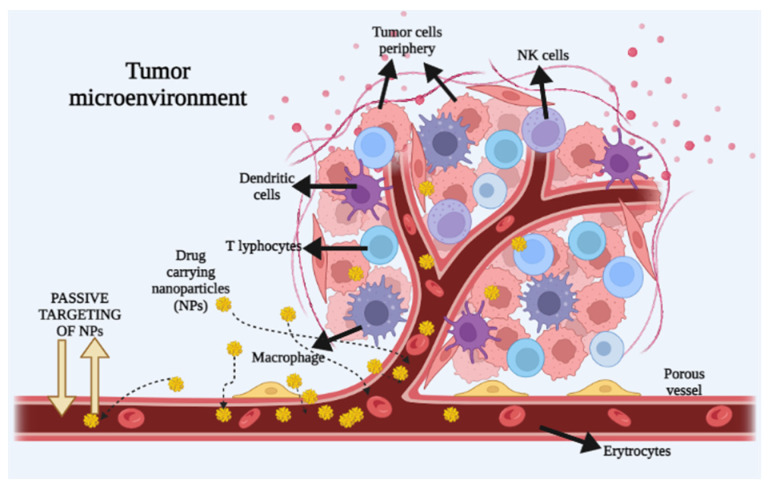
Schematic overview of tumor microenvironment and passive intratumor targeting of drug-carrying NPs. Due to the porous vessels, NPs can enter the vessels through passive diffusion and be delivered to the site of action in the tumor. This figure is similar those published before in studies by Cheng and Santos [135] and Roscigno et al. [136].

**Figure 4 ijms-24-12827-f004:**
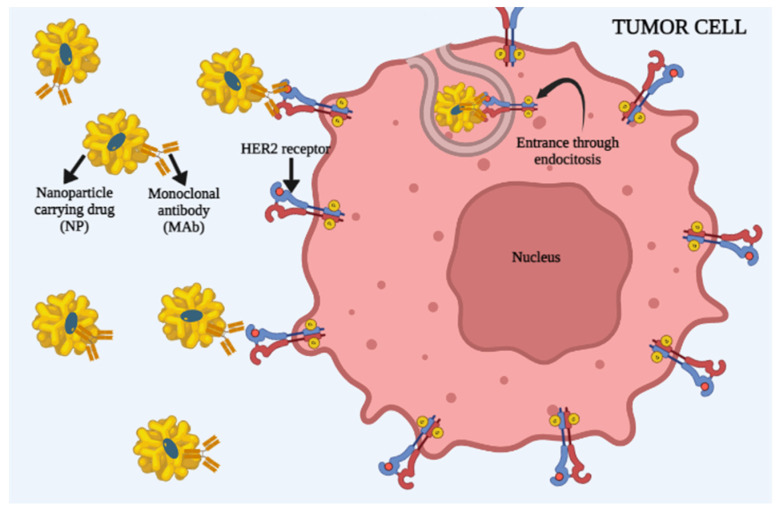
A schematic overview of active tumor cell targeting when the ligand on NPs carrying drug is a monoclonal antibody (MAb). When MAb binds with, in this case specifically, HER2 receptor on the tumor cell, it enters through the process of endocytosis. This enables NPs to release the drug that they are carrying inside the cell, which should lead to the apoptosis of the cell.

**Figure 5 ijms-24-12827-f005:**
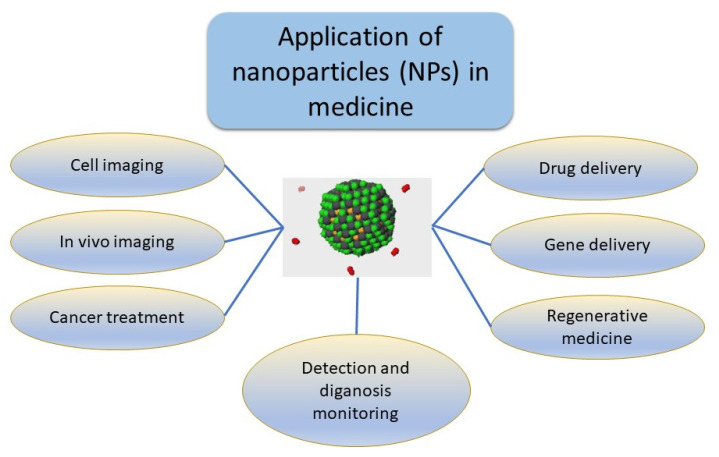
Schematic overview of current applications of NPs in medicine, which include cell imaging, in vivo imaging, drug delivery, gene delivery, cancer treatment and regenerative medicine.

**Table 1 ijms-24-12827-t001:** Approved inorganic nanomedicine drugs on the market for cancer treatment [20].

Product	Drug	Nanotechnology Platform	Cancer Type	Approval	Advantages	Toxicity
NANOTHERM	Fe_2_O_3_	Nanoparticles of superparamagnetic iron oxide coated with amino silane	Glioblastoma, prostate and pancreatic cancers	2013 (European medicine agency, EMA)	Upon interstitial administration, high blood circulation time and tumor uptake (EPR), heat production under stimulation with EMF and theranostic properties	Moderate adverse effect
NBTXR3 (HENSIFY)	Hafnium oxide nanoparticles stimulated with external radiation to enhance tumor cell death via electron production	Hafnium oxide nanoparticles	Locally advanced squamous cell carcinoma	2019 (CE Mark)	Radiotherapy enhancer	Injection site pain, hypotension and radiation skin injury

**Table 2 ijms-24-12827-t002:** Inorganic nanomedicine drugs under clinical trial for cancer treatment [20,21,22,23,24,25,26].

Name of Drug	Active Ingredients/Drugs Used	Nanocarrier/Formulation Type	Cancer Type	Properties/Objectives	Status
CARBON NANOPARTICLES	Carbon nanoparticle	Carbon nanoparticle	Advanced gastric cancer	Harvest lymph nodes after surgery	Phase 3
MAGNETIC NANOPARTICLES	Iron nanoparticle	Nanoparticle	Prostate cancer	Magnetic thermo-ablation	Early phase 1 (completed)
AGUIX GADOLINIUM-BASED NANOPARTICLES	AGuIX	Gadolinium-based nanoparticle	Centrally located lung tumors and pancreatic cancer	Safety and efficacy, stereotactic magnetic resonance-guided adaptive radiation therapy	Phase 1 & 2
MR-LINAC-SPION	Ferumoxytol	Iron oxide nanoparticles (SPION)	Primary and metastatic hepatic cancers	Radiotherapy	Not mentioned
CD24-GOLD NANOCOMPOSITE	CD24 primer and gold nanoparticle	Gold nanoparticles	Salivary gland tumors	Diagnostic tool, biomarker	Not mentioned
MAGNETIC PARTICLE-ICG	Magnetic tracers (FerroTrace) and indocyanine green (ICG)	Magnetic nanoparticles	Colorectal cancer	Feasibility of sentinel lymph node (SLN) mapping and safety	Phase 1 and 2
NANOTHERM®	Iron nanoparticles	Iron nanoparticles	Intermediate-risk prostate cancer	NanoTherm ablation	Not applicable
CARBON NANOPARTICLES	Carbon nanoparticles and indocyanine green	Carbon nanoparticles	Colorectal cancer	Lymph node tracers	Phase 2 and 3
NBTXR3	Hafnium oxide	Nanoparticles	Locally advanced or borderline-resectable pancreatic cancer	Particle activation by radiation therapy	Phase 1
SILICON INCORPORATED WITH QUATERNARY AMMONIUM POLYETHYLENIMINE NANOPARTICLES	Silicon	Quaternary ammonium poly-ethylenimine nanoparticles	Carcinoma of head and neck	Antibacterial activity	Phase 1
NBTXR3	Hafnium oxide	Hafnium oxide-containing nanoparticles	Esophageal cancer	Radiation therapy with concurrent chemotherapy	Phase 1
SILICA NANOPARTICLES	Silica nanoparticles	Silica nanoparticles	Head and neck melanoma	Bioimaging	Phase 1 and 2
POLYMERIC NANOPARTICLES	Polymeric nanoparticles	Polymeric nanoparticles	Colorectal cancer	Targeting somatostatin receptors	Phase 1
NANOPARTICLE	Inorganic nanoparticle	Nanoparticle	Advanced breast cancer	Pharmacokinetic profile	Phase 1
PLGA-PEG NANOPARTICLES	Amphiphilic polymer	PLGA nanoparticles	Squamous cell carcinoma	Therapeutic efficacy	Phase 2
SUPERPARAMAGNETIC IRON OXIDE NANOPARTICLE	SPIONs	Iron nanoparticle	Breast and colon cancer cells	Radio-sensitization of cancer cells Hyperthermia effect on cancer cells	Not mentioned
CRIPEC NANOPARTICLES	Cisplatin, carboplatin and oxaliplatin nanoparticles	Nanoparticle	Platinum resistant ovarian cancer	Chemotherapeutic eradication of cancer	Phase 2 (completed)

**Table 3 ijms-24-12827-t003:** FDA- or EMA-approved liposome-based drugs used in cancer treatment [20,21,22,23,24,25,26].

Clinical Products	Active Agent	Lipid/Lipid:Drug	Indication
DOXIL	Doxorubicin	HSPC:Cholesterol:PEG 2000-DSPE	Ovarian, breast cancer, Kaposi’s sarcoma
DAUNOXOME	Daunorubicin	DSPC and Cholesterol	AIDS-related Kaposi’s sarcoma
DEPOCYT	Cytarabine/Ara-C	DOPC, DPPG, Cholesterol and Triolein	Neoplastic meningitis
MYOCET	Doxorubicin	EPC:Cholesterol	Combination therapy with cyclophosphamide in metastatic breast cancer
MEPACT	Mifamurtide	DOPS:POPC	High-grade, resectable, non-metastatic osteosarcoma
MARQIBO	Vincristine	SM:Cholesterol	Acute lymphoblastic leukemia
ONIVYDE	Irinotecan	DSPC:MPEG-2000:DSPE	Combination therapy with fluorouracil and leucovorin in metastatic adenocarcinoma of the pancreas
VYXEOS/CPX-351	Cytarabine: daunorubicin	DSPG:DSPC:CL	Newly diagnosed therapy–related acute myeloid leukemia, acute myeloid leukemia
ZOLSKETIL	Doxorubicin	HSPC:CL:MPEG	Metastatic breast cancer, advanced ovarian cancer, multiple myeloma, AIDS-related Kaposi’s sarcoma
LIPO-DOX	Doxorubicin	DSPC:CL:MPEG	Kaposi’s sarcoma, ovarian cancer, breast cancer, multiple myeloma
LIPOPLATIN	Cisplatin	SPC-3:DPPG:CL:	Malignant pleural effusions
SPI-077	Cisplatin	HSPC:CL:MPEG2000-DSPE	Ovarian cancer

**Table 4 ijms-24-12827-t004:** List of drug candidates for cancer nano-therapy that are used in clinical practices and their types, listed in alphabetical order.

Drug	Type	Status in Clinical Practice
ABRAXANE®	Protein-bound paclitaxel, also known as nanoparticle albumin–bound	2005 FDA, 2008 EMA approval for solid cancers, 2012 and 2013 widened approvals
DAUNOXOME®	NonPEGylated daunorubicin citrate liposome injection	1996 FDA approval for HIV-related Kaposi’s sarcoma. The permanent discontinuation was purely a business decision.
DEPOCYT®	Liposomal cytarabine	1999 FDA approval for Lyposomatous meningitis 2017 Permanent discontinuation due to persistent technical issues in manufacturing process.
DOXIL®	Doxorubicin enclosed in uni-lamellar liposome coated with PEG	1995 FDA approved as first nanodrug used to treat metastatic ovarian cancer and AIDS-related Kaposi’s sarcoma
ELIGARD®	Leuprolide acetate	FDA approval for prostate cancer
GENEXOL-PM®	Paclitaxel-loaded polymeric micelle	2007 EMA and Korea for breast and lung cancer
LIPUSU®	(Liposomal paclitaxel)	2006 China for solid cancers
HENSIFY® (NBTXR3)	Radio-enhancer composed of hafnium oxide nanoparticles	FDA 2019 locally advanced soft tissue sarcoma
MARQIBO®	vincristine sulfate liposome injection	2012 FDA Philadelphia chromosome-negative acute lymphoblastic leukemia FDA withdraws approval clinical trial failed to verify the clinical benefit of the drug.
MEPACT®	muramyl tripeptide phosphatidylethanolamine, encapsulated into liposomes (L-MTP-PE).	FDA 2001 and EMA 2009 for osteosarcoma; FDA 2007 and EMA 2009 denied approval
MYOCET®	Liposome-encapsulated doxorubicin (nonPEGylated)	2000 EMA for breast cancer
NANOTHERM®	Magnetic nanoparticles of iron oxide implanted into the tumor or cavity wall and heated by alternating magnetic field	EMA 2010 and FDA 2018 for glioblastoma and prostate Cancer
ONCASPAR®	PEGasparaginase	FDA 1994 for acute lymphocytic leukemia
ONIVYDE®	Topoisomerase I inhibitor with irinotecan contained within a liposomal sphere.	FDA 2015 metastatic adenocarcinoma
ONTAC®	(Engineered protein combining interleukin-2 and diphtheria toxin)	Cutaneous T-cell lymphoma FDA (1999)
SMANCS®	Poly(styrene-co-maleic acid)-conjugated neo-carzinostatin	Hepatoma Japan (1997)
VYXEOS®	(Daunorubicin/cytarabine)fixed-dose chemotherapy combination	FDA 2017, EMA 2018 newly-diagnosed therapy-related acute myeloid leukemia (t-AML) or AML with myelodysplasia-related changes (AML-MRC) in people aged one year of age and older.

## Data Availability

Not applicable.

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
