# Peer review of "Nanoparticles in Medicine: Current Status in Cancer Treatment"

_ijms, 2023, doi:10.3390/ijms241612827_

Round 1

Reviewer 1 Report

The manuscript by D. Trivanovic and the team presents a review of the current status of nanoparticles in cancer management. The team discusses different types of nanoparticles ranging from organic to inorganic origin.  

Most of the contents of the manuscript are well-known in the literature, and no significant discussion was provided. 

The information in the figures is very general and not informative. 

Table 4 shows nanoparticles with combined gene delivery. What kind of genes were used for combinatorial therapy?  

The citation style of the manuscript is not consistent. 

The information in Table 3 and Table 4 is redundant. 

Author Response

Reviewer 1

Comments and Suggestions for Authors

The manuscript by D. Trivanovic and the team presents a review of the current status of nanoparticles in cancer management. The team discusses different types of nanoparticles ranging from organic to inorganic origin.  

Most of the contents of the manuscript are well-known in the literature, and no significant discussion was provided.

The information in the figures is very general and not informative. 

The main idea for the manuscript preparation was indeed to provide an overview of a very vast and highly diversified literature on nanoparticles applications in medicine. The manuscript gives comprehensive and basic relevant information for each nanoparticle type with references. The review was conceived as an introductory overview for scientist interested in the field without deep previous knowledge on certain nanoparticle types applications in cancer treatment.

Table 4 shows nanoparticles with combined gene delivery. What kind of genes were used for combinatorial therapy?

The Table 4 caption was unintentionally wrongly written. We have corrected the caption now into: “List of drug candidates for cancer nanotherapy that are used in clinical practices and their types, listed in alphabetical order”.

The citation style of the manuscript is not consistent.

We have carefully adjusted the citations and added new once as required by reviewer 2.

The information in Table 3 and Table 4 is redundant.

Table 3 gives a list of FDA- or EMA-approved liposome-based drugs used in cancer treatment while Table 4 gives specifically a list of drug candidates for cancer nanotehrapy that are used in clinical practices and their types. Some information may be overlapping but two tables are given for a specific reason to comprehensively enlist certain product group for the reader.

Reviewer 2 Report

The reviewer recommends a revision by a native speaker. 

Overall, the reviewer recommends focusing on in vivo experiments. While in vitro experiments have their usefulness, for the purpose of real translation, they hold less significance.

The reviewer recommends a change in reference 1. I believe that using DOI:10.3322/caac.21660 would be more accurate.

The sentence "The main cancer treatments include surgery, radiotherapy, chemotherapy, hormone therapy, 69 immunotherapy or combinations of these therapies." needs references.

The sentence "Some of these treatment modalities, such as, for exam-70 ple, chemotherapy, lack specificity and have problems with cytotoxicity, stem-like cell growth and multi-71 drug resistance." needs references.

The paragraph "Through the use of nanoparticles (NPs), indeed, some important 77 goals in cancer therapy may be achieved: decreased adverse effects of delivered drugs, the possibility to 78 prepare a myriad of nanoformulations for drug delivery, and the targeting of tumoral cells and their de-79 struction with the help of NPs’ electrical, magnetic or optical characteristics." needs references

Authors must describe in the sentence "Indeed, the property of na-80 nosized materials of accumulating in tumor tissues due to specific changes in the tumor vasculature had 81 been discovered decades ago5" the EPR effects, and Pf. Maeda may be named. Moreover, the reviewer recommends discussing the great controversy surrounding the EPR effect. Authors can refer to the pioneering work of Pf. Joseph W. Nichols (University of Utah), Prof. Warren Chan (especially his works of 2016), and Dr. Carlos Caro (IBIMA-Plataforma BIONAND). Furthermore, the reviewer recommends including a brief description of the differences between cell targeting and tumor targeting, as they are distinct concepts. This description can be added either in this section or elsewhere in the manuscript.

The sentence "Some of the best studied 85 NPs include metallic gold and silver NPs, quantum dots (QDs), polymeric NPs, carbon nanotubes and 86 graphene." needs references

The sentence "Inorganic NPs are generally biocompatible and highly stable compared to organic materials." since dendrimers, for instance, are extremly stable.

Table 1. Are the authors certain that Nanotherm is administered intravenously? If the authors visit the MagForce page, they can observe that it is administered stereotactically.

Authors describe "In general, QDs 117 generate intracellular reactive oxygen species (ROS), thereby causing cancer cell death through oxidative 118 DNA damage 20, or directly impacting the immunological processes through the enhancement of proin-119 flammatory signaling through different immune modulators including cytokines, chemokines and metal-120 loproteinases 21. This is why in vivo applications of QDs have an interesting potential for cancer therapy, 121 either by the use of QDs alone or using QDs conjugated with anticancer drugs 22." However, the activation of the immune system can lead to side effects that, in extreme cases, may even result in patient death. Furthermore, the majority of the results discussed by the authors in this section are related to in vitro experiments, which do not involve the immune system. The reviewer recommends a summary table at the end of the sections. Authors must add information about in vitro/in vivo experiments, administration route...

Could the authors please explain why quantum dots (QDs) are not considered metallic nanoparticles (NPs)? It is possible that the authors want to refer Section 2.1.2, as plasmonic nanoparticles (NPs). Also magnetic NPs are metallic NPs. Indeed, check line 143, with information about magnetic properties.

In the sentence "Importantly, the metallic NPs exhibit increased stability and 145 half-life in circulation, biodistribution and specific targeting into the target site, which is relevant for their 146 clinical application." How stability is increased? by surface functionalization? By encapsulation in organic NPs?

The sentence "Due to their rela-148 tively small size, metallic NPs can travel by passive transport through the junctions of loose cells specific to 149 tumor tissue, thus causing their increased accumulation at the tumor site 31." provide repeated information (EPR effects)

Authors describe "Some of the metallic NPs tested in clinical applications include silver (AgNPs), gold 152 (AuNPs), palladium, titanium, zinc, and copper-based NPs33. For example, Barabadi et al. reviewed all the 153 published data for AgNPs’ effects on in vitro lung cancer cell lines" but clinical applications is completly different from in vitro experiments.

The sentence "Inorganic porous nanomaterials have been acknowledged as promising drug carriers due to particular 169 structural properties such as high loading capacity of biomolecules, possibilities of various surface modifi-170 cations and controllable release of drug molecules." needs references

The paragraph "The majority of porous 172 materials are made of randomly oriented and repeating single units forming pores. Pores are thus used for 173 the embedding of desired molecules, i.e., drugs or antibodies." needs references

The sentence "A large group of porous materials are zeolites, crystal-176 line aluminosilicates with a porous structure on the micro- and nano-scale that occur naturally but can also 177 be prepared in controlled reactions to obtain materials with controlled physico-chemical properties and 178 pore size." needs references

The sentence "The single unit, the building block of a zeolite material, is a tetrahedron made of atoms (like Si and 179 Al) that are bound together through oxygen atoms in between two tetrahedron units." needs references

The sentence "Zeolites have, accordingly, been recently prepared as nanosized materials to advance their usage in indus-183 trial applications." needs references

Section 2.1.3. Few in vivo experiments are described. Please check my second comment.

The sencente "Magnetic NPs are studied for cancer therapy applications due to their potential in targeted drug delivery." needs references. Moreover, the authors must describe the applications of magnetic hyperthermia, which is also a crucial application of MNPs..

The sentence "One of the main approaches is to develop porous microspheres from magnetic spinel ferrites MxFe3−xO4 (M 219 = Fe, Zn)." needs references. However, it is out of the scope of the review. Microparticles are completely different from nanoparticles

The sentences "Due to the high magnetism, these magnetic nanoparticles can be easily manipulated by a magnet 220 within the vascular system and, if needed, they remain accumulated in a specific place within the organ-221 ism." needs references. However, authors must exercise caution regarding this information. Superparamagnetic nanoparticles cannot be attracted by a magnet, unlike ferromagnetic nanoparticles. Furthermore, there is a high risk of thrombosis following the intravenous administration of ferromagnetic nanoparticles, which could potentially lead to patient mortality.

Section 2.1.4 is not well organized. Please rewrite it.

The sentences "These NPs have broad application 268 in medicine due to their exceptional physical and chemical properties and acceptable levels of biocompati-269 bility." needs references. Indeed, there have been numerous concerns regarding the toxicity of carbon nanomaterials. It is advisable to thoroughly investigate and review the available literature on this topic to provide accurate and up-to-date information.

The paragraph "Gra-274 phene is, for example, a highly researched material in the area of drug design and drug delivery, but its use 275 in medical applications remains limited due to its hydrophobic nature. In contrast, graphene derivatives, 276 such as, for example, graphene oxide, show better properties for in vivo applications." needs references

The sentence "These nanoparticles are formed of 60 car-291 bon atoms, where the carbon atoms form a hollow sphere or ellipse that can easily accommodate drugs in 292 the hollow space." needs references

The paragraph "Even though inorganic nanoparticles have gained substantial attention in the area of cancer therapy, their 309 toxicity and the limitations for safe in vivo usage have prompted a high level of interest in the study of or-310 ganic, biocompatible nanoparticles for different cancer treatment applications as well. Such organic nano-311 particles are developed in such a way as to be nontoxic for cells and biodegradable and to not induce cellu-312 lar or tissue damage." needs references.

The sentence "Polymeric nanoparticles are highly stable, colloidal, biocompatible and biodegradable nanomaterials that 318 encapsulate hydrophobic chemotherapeutics in their matrices." needs references

Section 2.2.1. Could authors explain the diferences between chitosan NPs and Chitosan based nanogels? 

The paragraph "Still, liposomes may be unstable in vivo as some types of bilayers undergo oxida-364 tion or hydrolysis, but these issues may be overcome by careful choice of the material used for liposome 365 production, by a directed and controlled production process and by delivery of the formulation in the ly-366 ophilized form." needs references

Section 3.1 Passive intracellular transport...Do authors refers to Passive tumor targeting? AS I mentioned previously, cell targeting is completly different from tumor targeting.

Sentence 385. Authors describe here for the first time the EPR effect. But, as I mentioned previously, they described it (without named it)

The paragraph "In tumor tissue, the endothelium of blood vessels becomes more permeable than when in a normal, healthy 412 state. This happens because of hypoxia, where quickly growing tumors recruit new vessels or overflow the 413 existing ones, and those vessels are leaky and ensure passive transport of a wider range of small particles 414 entering the cell. Due to the misfunction of normal lymphatic drainage, NPs can stay inside the tumor and 415 its tissue much longer, while small molecule drugs, which are known to have a short circulation time, are 416 washed out first." must be moved to the EPR definition. Same for "Unfortunately, this passive entrance is highly dependent on the intrinsic tumor biology, which includes the 429 stage of angiogenesis and lymphangiogenesis, perivascular tumor growth and intratumor pressure109. An-430"

The sentence "NPs do not reach those that are poorly devel-432 oped, because of the lack of nutrient and oxygen supply." needs references

The paragraph "way to bypass these problems is to apply 437 enhanced permeability and retention (EPR) mechanism enhancers such as bradykinin, nitric oxide, peroxy-438 nitrite, prostaglandins, vascular permeability factor (VPF) and/or vascular endothelial growth factor 439 (VEGF), and other cytokines or macromolecules111. These enhancers induce hypertension and/or vessel 440 normalization, which could possibly enhance tumor overflow. Besides EPR enhancers, there are other ap-441 proaches such as radiation, ultrasound, hyperthermia or photoimmunotherapy, which can also increase 442 NPs’ infusion, as they allow for tumor-selective combination therapy by guided physical approaches such 443 as, for example, multimodal-imaging-guided tumor inhibition." needs references. In addition, authors must move to EPR limitations and oportunities.

Section 3.2 Active intracellular transport...Do authors refers to Active tumor targeting?  EPR effect is primarily observed in vivo, within the context of tumor vasculature and tumor tissues. It is not applicable or relevant to cell culture experiments.

The paragraph "Binding affinity is defined as the strength of a molecule 463 to bind with a targeted counterpart molecule. An example of increased binding affinity is the NPs that have 464 more folates on their surface; however, this has a limit. Binding affinities can also be decreased due to a 465 very high concentration of ligands on the NP’s surface. The reason for that is that there are many steric 466 binding interferences that ultimately prevent the binding of a ligand to the antigen." needs references

In fact, it is recommended to rewrite Section 3.2 due to the presence of confusing information.

The apragraph "A specific problem is the biological barriers 514 such as the tumor microenvironment (TME), as in pancreatic ductal adenocarcinoma, or the blood–brain 515 barrier (BBB). The TME is a biologically functional barrier that uses a system composed of abnormal vas-516 culature, fibroblasts and various immune cells, all fixed in an extracellular matrix and with a pressure gra-517 dient between the interstitial space and a tumor mass. Tumor cells in the core become hypoxic and the an-518 oxic metabolic pathway leads to a decrease in pH status." needs references. Moreover BBB has been previously described. TME must be described in the introduction, since is a key factor in NPs accumulation.

Section 5 is confusing. There is lot of repeated information. In fact, Which is the diference between table 3 and table 4? Moreover, there is information about sRNA, vaccines... 

The paragraph "Accordingly, nanoparticles are now tested in clinical trials as vehicles or drug combinations for 579 immunotherapeutic and immunomodulatory agents in the form of vaccines, cytokines and adoptive cellu-580 lar therapies." needs references.

The sentence "Radiation can be extremely toxic, not only to cancer cells but to normal, healthy cells, 586 which dramatically limits its use." needs references

The paragraph "Combined nanotechnology and radiotherapy effects rely on the interac-587 tion between X-rays and nanoparticles due to the inherent atomic-level properties of the materials. The first 588 mechanism of the interaction between X-rays and NPs includes the NPs with high atomic number Z that 589 enhance the photoelectric and Compton effects, which leads to the emission of secondary electrons that add 590 to tumor cell destruction, to increase the efficacy of conventional radiation therapy." needs references

The sentence "In short, classical physics, including the Laplace 656 equation and Epstein–Plesset theory, states that nanobubbles, due to their small size, should not exist, and 657 gas introduced into liquid will either dissolve or create microbubbles and macrobubbles that rise to the top 658 of the vessel." needs references

The sentence "Nanobubbles adsorb hydroxyl ions from solutions and, in part 660 due to that surface charge, are stable for weeks or even months."

Why authors describe Hydrogen molecules?

Section 6.2 Nanobubbles characterization is out of the scope of the review.

Authors described that Electrolyzed water can allevi-707 ate the side effects of chemotherapy and radiation. It is out of the scope of the review.

Section 6.5 have lot of repeated information. Lipids were previoussly described.

Section 6.6 There is some problem with references.

The quality of the English must be improved since the readability is poor sometimes

Author Response

Reviewer 2

Comments and Suggestions for Authors

The reviewer recommends a revision by a native speaker.

The manuscript has been revised by a professional MDPI language expert. The confirmation is uploaded in MDPI system.

We have also rewritten and deleted all the parts as per your kind suggestions. The parts that are now changed now are denoted in red font.

Overall, the reviewer recommends focusing on in vivo experiments. While in vitro experiments have their usefulness, for the purpose of real translation, they hold less significance.

We agree that the in vivo experiments have a true translational value, and have thus limited the in vitro studies in the paper just to those relevant for early phases of development of nanoparticles as they contribute to understanding of the molecular mechanisms of action. We would like accordingly to keep them in the text.

The reviewer recommends a change in reference 1. I believe that using DOI:10.3322/caac.21660 would be more accurate.

Thank you! This has been corrected.

The sentence "The main cancer treatments include surgery, radiotherapy, chemotherapy, hormone therapy, 69 immunotherapy or combinations of these therapies." needs references.

This is common clinical knowledge. We added now a link to the NIH page where this is also stated (https://www.cancer.gov/about-cancer/treatment )

The sentence "Some of these treatment modalities, such as, for exam-70 ple, chemotherapy, lack specificity and have problems with cytotoxicity, stem-like cell growth and multi-71 drug resistance." needs references.

This relates to references 2-4 and we now added them in the required position.

The paragraph "Through the use of nanoparticles (NPs), indeed, some important 77 goals in cancer therapy may be achieved: decreased adverse effects of delivered drugs, the possibility to 78 prepare a myriad of nanoformulations for drug delivery, and the targeting of tumoral cells and their de-79 struction with the help of NPs’ electrical, magnetic or optical characteristics." needs references

This relates to references 5-7 and we now added them in the required position.

Authors must describe in the sentence "Indeed, the property of na-80 nosized materials of accumulating in tumor tissues due to specific changes in the tumor vasculature had 81 been discovered decades ago5" the EPR effects, and Pf. Maeda may be named. Moreover, the reviewer recommends discussing the great controversy surrounding the EPR effect. Authors can refer to the pioneering work of Pf. Joseph W. Nichols (University of Utah), Prof. Warren Chan (especially his works of 2016), and Dr. Carlos Caro (IBIMA-Plataforma BIONAND). Furthermore, the reviewer recommends including a brief description of the differences between cell targeting and tumor targeting, as they are distinct concepts. This description can be added either in this section or elsewhere in the manuscript.

Thank you for this suggestions, we have included the required data along with references in this paragraph now.

The sentence "Some of the best studied 85 NPs include metallic gold and silver NPs, quantum dots (QDs), polymeric NPs, carbon nanotubes and 86 graphene." needs references

This is a general sentence that we derived from literature search and followed by examples in our manuscript. We have now rephrased it accordingly.

The sentence "Inorganic NPs are generally biocompatible and highly stable compared to organic materials." since dendrimers, for instance, are extremly stable.

Table 1. Are the authors certain that Nanotherm is administered intravenously? If the authors visit the MagForce page, they can observe that it is administered stereotactically.

We did not write that Nanotherm is administered intravenously. We are aware of its modality of application. Maybe this may be deduced from the explanation in the table which states “high blood circulation”. To make it clear, we have now added the general administration modality as well.

Authors describe "In general, QDs 117 generate intracellular reactive oxygen species (ROS), thereby causing cancer cell death through oxidative 118 DNA damage 20, or directly impacting the immunological processes through the enhancement of proin-119 flammatory signaling through different immune modulators including cytokines, chemokines and metal-120 loproteinases 21. This is why in vivo applications of QDs have an interesting potential for cancer therapy, 121 either by the use of QDs alone or using QDs conjugated with anticancer drugs 22." However, the activation of the immune system can lead to side effects that, in extreme cases, may even result in patient death. Furthermore, the majority of the results discussed by the authors in this section are related to in vitro experiments, which do not involve the immune system. The reviewer recommends a summary table at the end of the sections. Authors must add information about in vitro/in vivo experiments, administration route...

We have now emphasized these points raised by the Reviewer. As the provided examples in the text are from very preliminary studies, mainly in vitro studies, we have also emphasized this point.

Could the authors please explain why quantum dots (QDs) are not considered metallic nanoparticles (NPs)? It is possible that the authors want to refer Section 2.1.2, as plasmonic nanoparticles (NPs). Also magnetic NPs are metallic NPs. Indeed, check line 143, with information about magnetic properties.

We used the usual nomenclature from the literature as we cited it. For the QDs we have now however, added an additional paragraph to explain their unique physicochemical properties.

In the sentence "Importantly, the metallic NPs exhibit increased stability and 145 half-life in circulation, biodistribution and specific targeting into the target site, which is relevant for their 146 clinical application." How stability is increased? by surface functionalization? By encapsulation in organic NPs?

This has been explained now with addition of a new reference.

The sentence "Due to their rela-148 tively small size, metallic NPs can travel by passive transport through the junctions of loose cells specific to 149 tumor tissue, thus causing their increased accumulation at the tumor site 31." provide repeated information (EPR effects)

This has been removed now accordingly.

Authors describe "Some of the metallic NPs tested in clinical applications include silver (AgNPs), gold 152 (AuNPs), palladium, titanium, zinc, and copper-based NPs33. For example, Barabadi et al. reviewed all the 153 published data for AgNPs’ effects on in vitro lung cancer cell lines" but clinical applications is completly different from in vitro experiments.

This is true, we have now corrected the text accordingly.

The sentence "Inorganic porous nanomaterials have been acknowledged as promising drug carriers due to particular 169 structural properties such as high loading capacity of biomolecules, possibilities of various surface modifi-170 cations and controllable release of drug molecules." needs references

The reference has been added now.

The paragraph "The majority of porous 172 materials are made of randomly oriented and repeating single units forming pores. Pores are thus used for 173 the embedding of desired molecules, i.e., drugs or antibodies." needs references

The reference has been added now.

The sentence "A large group of porous materials are zeolites, crystal-176 line aluminosilicates with a porous structure on the micro- and nano-scale that occur naturally but can also 177 be prepared in controlled reactions to obtain materials with controlled physico-chemical properties and 178 pore size." needs references

The reference has been added now.

The sentence "The single unit, the building block of a zeolite material, is a tetrahedron made of atoms (like Si and 179 Al) that are bound together through oxygen atoms in between two tetrahedron units." needs references

The reference has been added now.

The sentence "Zeolites have, accordingly, been recently prepared as nanosized materials to advance their usage in indus-183 trial applications." needs references

The reference has been added now.

Section 2.1.3. Few in vivo experiments are described. Please check my second comment.

We agree that the in vivo experiments have a true translational value, and have thus limited the in vitro studies in the paper just to those relevant for early phases of development of nanoparticles as they contribute to understanding of the molecular mechanisms of action. We would like accordingly to keep them in the text.

The sentence "Magnetic NPs are studied for cancer therapy applications due to their potential in targeted drug delivery." needs references. Moreover, the authors must describe the applications of magnetic hyperthermia, which is also a crucial application of MNPs.

The reference has been added now.

The sentence "One of the main approaches is to develop porous microspheres from magnetic spinel ferrites MxFe3−xO4 (M 219 = Fe, Zn)." needs references. However, it is out of the scope of the review. Microparticles are completely different from nanoparticles

We agree that the sentence may be deleted and we removed it accordingly.

The sentences "Due to the high magnetism, these magnetic nanoparticles can be easily manipulated by a magnet 220 within the vascular system and, if needed, they remain accumulated in a specific place within the organ-221 ism." needs references.

The reference is given immediately after the next sentence.

However, authors must exercise caution regarding this information. Superparamagnetic nanoparticles cannot be attracted by a magnet, unlike ferromagnetic nanoparticles. Furthermore, there is a high risk of thrombosis following the intravenous administration of ferromagnetic nanoparticles, which could potentially lead to patient mortality.

We have now rewritten this part to make a clear statement on supramagnetic particles and their potential toxicity.

Section 2.1.4 is not well organized. Please rewrite it.

The section has now been reorganized.

The sentences "These NPs have broad application 268 in medicine due to their exceptional physical and chemical properties and acceptable levels of biocompati-269 bility." needs references. Indeed, there have been numerous concerns regarding the toxicity of carbon nanomaterials. It is advisable to thoroughly investigate and review the available literature on this topic to provide accurate and up-to-date information.

Thank you, we are aware of this and the sentence has been rephrased accordingly. The original idea was to emphasize the properties for translational potential. We also added a new reference now.

The paragraph "Gra-274 phene is, for example, a highly researched material in the area of drug design and drug delivery, but its use 275 in medical applications remains limited due to its hydrophobic nature. In contrast, graphene derivatives, 276 such as, for example, graphene oxide, show better properties for in vivo applications." needs references

The reference on both statements has been added now.

The sentence "These nanoparticles are formed of 60 car-291 bon atoms, where the carbon atoms form a hollow sphere or ellipse that can easily accommodate drugs in 292 the hollow space." needs references

The reference has been added now.

The paragraph "Even though inorganic nanoparticles have gained substantial attention in the area of cancer therapy, their 309 toxicity and the limitations for safe in vivo usage have prompted a high level of interest in the study of or-310 ganic, biocompatible nanoparticles for different cancer treatment applications as well. Such organic nano-311 particles are developed in such a way as to be nontoxic for cells and biodegradable and to not induce cellu-312 lar or tissue damage." needs references.

The reference has been added now.

The sentence "Polymeric nanoparticles are highly stable, colloidal, biocompatible and biodegradable nanomaterials that 318 encapsulate hydrophobic chemotherapeutics in their matrices." needs references

The reference has been added now.

Section 2.2.1. Could authors explain the diferences between chitosan NPs and Chitosan based nanogels?

Nanogel is one type of formulation particularly relevant for topical applications and has thus been discussed as a formulation, including the chitosan-based formulation mentioned in this appropriate section. Now we how slightly changed the text to make this point for the reader as well.

The paragraph "Still, liposomes may be unstable in vivo as some types of bilayers undergo oxida-364 tion or hydrolysis, but these issues may be overcome by careful choice of the material used for liposome 365 production, by a directed and controlled production process and by delivery of the formulation in the ly-366 ophilized form." needs references

The reference has been added now.

Section 3.1 Passive intracellular transport...Do authors refers to Passive tumor targeting? AS I mentioned previously, cell targeting is completly different from tumor targeting.

Thank you, we accepted the suggestion to clearly distinguish between these two targeting types and have rephrased accordingly the text in our manuscript where applicable.

Sentence 385. Authors describe here for the first time the EPR effect. But, as I mentioned previously, they described it (without named it)

We have described now the EPR previously as suggested and here we elaborate it in more details.

The paragraph "In tumor tissue, the endothelium of blood vessels becomes more permeable than when in a normal, healthy 412 state. This happens because of hypoxia, where quickly growing tumors recruit new vessels or overflow the 413 existing ones, and those vessels are leaky and ensure passive transport of a wider range of small particles 414 entering the cell. Due to the misfunction of normal lymphatic drainage, NPs can stay inside the tumor and 415 its tissue much longer, while small molecule drugs, which are known to have a short circulation time, are 416 washed out first." must be moved to the EPR definition. Same for "Unfortunately, this passive entrance is highly dependent on the intrinsic tumor biology, which includes the 429 stage of angiogenesis and lymphangiogenesis, perivascular tumor growth and intratumor pressure109. An-430"

This has been removed now and the paragraph adjusted accordingly.

The sentence "NPs do not reach those that are poorly devel-432 oped, because of the lack of nutrient and oxygen supply." needs references

The reference has been added now.

The paragraph "way to bypass these problems is to apply 437 enhanced permeability and retention (EPR) mechanism enhancers such as bradykinin, nitric oxide, peroxy-438 nitrite, prostaglandins, vascular permeability factor (VPF) and/or vascular endothelial growth factor 439 (VEGF), and other cytokines or macromolecules111. These enhancers induce hypertension and/or vessel 440 normalization, which could possibly enhance tumor overflow. Besides EPR enhancers, there are other ap-441 proaches such as radiation, ultrasound, hyperthermia or photoimmunotherapy, which can also increase 442 NPs’ infusion, as they allow for tumor-selective combination therapy by guided physical approaches such 443 as, for example, multimodal-imaging-guided tumor inhibition." needs references. In addition, authors must move to EPR limitations and oportunities.

The reference stays for the whole paragraph at the end of it. This strategies for overcoming the EPR limitations should remain in this paragraph as it deals specifically with the passive NP tumor delivery.

Section 3.2 Active intracellular transport...Do authors refers to Active tumor targeting?  EPR effect is primarily observed in vivo, within the context of tumor vasculature and tumor tissues. It is not applicable or relevant to cell culture experiments.

We described herein specific cellular targeting and we rephrased the title.

The paragraph "Binding affinity is defined as the strength of a molecule 463 to bind with a targeted counterpart molecule. An example of increased binding affinity is the NPs that have 464 more folates on their surface; however, this has a limit. Binding affinities can also be decreased due to a 465 very high concentration of ligands on the NP’s surface. The reason for that is that there are many steric 466 binding interferences that ultimately prevent the binding of a ligand to the antigen." needs references

The reference stays for the whole paragraph at the end of it.

In fact, it is recommended to rewrite Section 3.2 due to the presence of confusing information.

We have stated clearly now that we described cell targeting and rephrased the corresponding text in the paragraph 3.2.

The apragraph "A specific problem is the biological barriers 514 such as the tumor microenvironment (TME), as in pancreatic ductal adenocarcinoma, or the blood–brain 515 barrier (BBB). The TME is a biologically functional barrier that uses a system composed of abnormal vas-516 culature, fibroblasts and various immune cells, all fixed in an extracellular matrix and with a pressure gra-517 dient between the interstitial space and a tumor mass. Tumor cells in the core become hypoxic and the an-518 oxic metabolic pathway leads to a decrease in pH status." needs references. Moreover BBB has been previously described. TME must be described in the introduction, since is a key factor in NPs accumulation.

We have removed this section to the introductory section where we now describe the EPR effect.

Section 5 is confusing. There is lot of repeated information. In fact, Which is the diference between table 3 and table 4? Moreover, there is information about sRNA, vaccines... 

Table 3 gives a list of FDA- or EMA-approved liposome-based drugs used in cancer treatment while Table 4 gives specifically a list of drug candidates for cancer nanotehrapy that are used in clinical practices and their types. Some information may be overlapping but two tables are given for a specific reason to comprehensively enlist certain product group for the reader.

The paragraph "Accordingly, nanoparticles are now tested in clinical trials as vehicles or drug combinations for 579 immunotherapeutic and immunomodulatory agents in the form of vaccines, cytokines and adoptive cellu-580 lar therapies." needs references.

The references have been added now.

The sentence "Radiation can be extremely toxic, not only to cancer cells but to normal, healthy cells, 586 which dramatically limits its use." needs references

The reference has been added now.

The paragraph "Combined nanotechnology and radiotherapy effects rely on the interac-587 tion between X-rays and nanoparticles due to the inherent atomic-level properties of the materials. The first 588 mechanism of the interaction between X-rays and NPs includes the NPs with high atomic number Z that 589 enhance the photoelectric and Compton effects, which leads to the emission of secondary electrons that add 590 to tumor cell destruction, to increase the efficacy of conventional radiation therapy." needs references

The reference has been added now.

The sentence "In short, classical physics, including the Laplace 656 equation and Epstein–Plesset theory, states that nanobubbles, due to their small size, should not exist, and 657 gas introduced into liquid will either dissolve or create microbubbles and macrobubbles that rise to the top 658 of the vessel." needs references

The references are added now.

The sentence "Nanobubbles adsorb hydroxyl ions from solutions and, in part 660 due to that surface charge, are stable for weeks or even months."

Why authors describe Hydrogen molecules?

We describe nanobubbles of hydrogen and elaborate on their potential role in cancer nanotreatment in the paragraph.

Section 6.2 Nanobubbles characterization is out of the scope of the review.

We have removed the paragraph accordingly.

Authors described that Electrolyzed water can allevi-707 ate the side effects of chemotherapy and radiation. It is out of the scope of the review.

We have removed the statement accordingly.

Section 6.5 have lot of repeated information. Lipids were previoussly described.

Section 6.6 There is some problem with references.

We have now combined and shortened the sections 6.5. and 6.6. accordingly.

Round 2

Reviewer 1 Report

The manuscript by D. Trivanovic and the team presents a fundamental review of the current status of nanoparticles in cancer management. The group discusses different types of nanoparticles ranging from organic to inorganic origin.  

Overall, the writing and discussion quality is not up to the mark of IJMS standards. 

The authors provided very general information about nanoparticles, and all of the figures are not informative and do not contain new information or new direction from the author's perspective. 

Though the authors provide essential scientific information, the direction and story of the manuscript is not clear and focused. 

Line 96: specific changes in the tumor vasculature. What are those specific changes? The authors must provide clear thinking. 

Line 97: the blood–brain barrier (BBB) is a barrier to delivering therapeutics to the brain. HOW is BBB a specific problem in the NP delivery of tumors? Blood-brain-tumor-barrier (BBTB)?

This manuscript requires extensive simple grammatical checks.  

For example, Line 90: This controversies.   The correct notion is this controversy or these controversies.

Author Response

The manuscript by D. Trivanovic and the team presents a fundamental review of the current status of nanoparticles in cancer management. The group discusses different types of nanoparticles ranging from organic to inorganic origin.

Overall, the writing and discussion quality is not up to the mark of IJMS standards. 

The authors provided very general information about nanoparticles, and all of the figures are not informative and do not contain new information or new direction from the author's perspective. 

Though the authors provide essential scientific information, the direction and story of the manuscript is not clear and focused.

We would not like to argue with personal opinions of the reviewer which are legitimate but should not stay for an objective evaluation of the value of our manuscript. As we previously explained, the main idea for the manuscript preparation was to provide an overview of a very vast and highly diversified literature in one single paper on nanoparticles applications in medicine and tumour treatment. The manuscript gives accordingly, a comprehensive and relevant information for each nanoparticle type with references. The review was conceived as an introductory overview for scientist interested in the field without deep previous knowledge on certain nanoparticle types applications in cancer treatment. This is why the text is focused in its parts on certain topics separately.

Line 96: specific changes in the tumor vasculature. What are those specific changes? The authors must provide clear thinking. 

This is already explained in the parenthesis:

(…) “Indeed, the property of nanosized materials to accumulate in tumor tissues due to specific changes in the tumor vasculature (enhanced permeability and retention effect, EPR), had been discovered decades ago by Maeda and Matsumura….“ (…)

Line 97: the blood–brain barrier (BBB) is a barrier to delivering therapeutics to the brain. HOW is BBB a specific problem in the NP delivery of tumors? Blood-brain-tumor-barrier (BBTB)?

Thank you for your observations but the blood brain barrier is present in tumor patients as well and the abbreviation should stay as BBB.

Comments on the Quality of English Language

This manuscript requires extensive simple grammatical checks.  

For example, Line 90: This controversies.  à The correct notion is this controversy or these controversies.

Thank you very much for the kind comment. Our manuscript has been edited by a language expert previously, except the new added sentences. We have now carefully revised them as suggested.

Reviewer 2 Report

The authors have addressed all the requirements provided by the reviewer.

Author Response

Thank you very much for your suggestions that increased the quality of our manuscript.

Round 3

Reviewer 1 Report

The manuscript by D. Trivanovic and the team presents a fundamental review of the current status of nanoparticles in cancer management. The group discusses different types of nanoparticles ranging from organic to inorganic origin. We recommend a detailed revision addressing the following issues carefully to reach more audiences and readers of different disciplines before considering a possible publication.

1.    The authors claimed that the manuscript had been edited by a language expert. However, still, English errors were observed.  For example: A specific problem on NPs delivery to tumors, are  A specific problem in NPs delivery to tumors is

2.    Line 96: specific changes in the tumor vasculature. Please do immediately mention those specific changes. The readers will not wait for the suspense to be revealed in the next sentences about those specific changes. For example, nanosized materials accumulate in tumor tissues due to leakyvasculature and underdeveloped lymphatic drainage in the tumor region (enhanced permeability and retention effect, EPR) specific changes in the tumor vasculature.

3.    The prefix of dextrorotatory or levorotatory isomer should be in small capital letters. For example, Figure 2: N-acetyl-D-glucosamine  N-acetyl-d-glucosamine and D-glucosamine d-glucosamine 

Follow the following instructions for small capital letters. Type small case letters l or d  Select the small case letter  Go to Format  Go to Font  Select small caps shown in “effects.”     

4.    Tumor microenvironment (TME) is a barrier in pancreatic ductal adenocarcinoma. Then the authors should mention the blood-brain barrier (BBB) in what kind of tumors. For example, barriers such as the tumor microenvironment (TME) in pancreatic ductal adenocarcinoma, or the blood–brain barrier (BBB)in brain tumors. 

5.    The authors stated that Figure 2. Examples of polymer-based nanoparticles. The shown chemical structures are polymer-based nanoparticles? They are polymers. Not nanoparticles. Please change Examples of polymers used for preparing polymer-based nanoparticles

6.    Some words were used many times throughout the manuscript. They will increase the word count but not the scientific information. Show the abbreviation or acronym only once after its appearance in the manuscript. For example, Lines 247, 268, 280, 354, 417. poly (ethylene glycol).

Lines 97, 364, 698 brain–blood barrier (BBB). 

Lines 384, 417, Figure 2, poly(lactic-co-glycolic acid) (PLGA) poly(lactic-co-glycolic acid) (PLGA) and poly (lactic-co-glycolide) (PLG) are same. Why two different terminology? Use consistent terminology throughout the manuscript. 

7.    No special acronyms or abbreviations are required if the information is given only once. They simply increase the word count but not special scientific information.

For example, Line 103, high-intensity focused ultrasound (HIFU) 

Line 416: poly (aspartic acid) (PAA), 

Line 417: poly(caprolactone) PCL 

Line 487: dissociation constant (KD).

Line 517: (Tf-Mpeg-pe) 

Line 750: (TCGA) 

Line 749: NIH

8.    Some acronyms or abbreviations were not expanded. All readers cannot immediately follow such acronyms or abbreviations. 

Lines 509 and 526: HER2  human epidermal growth factor receptor 2 (HER2)

Lines 624, 626, and 648: CRISPR/cas9  clustered regularly interspersed short palindromic repeats (CRISPR)/CRISPR-associated protein 9 (Cas9)

Lines 684 and 694: IGF-1  Insulin-like growth factor 1 (IGF-1)

Line 693: Nrf2  Nuclear factor erythroid 2-related factor 2

Many more were found. Please address all of them.  

9.    Table 4: DOXIL®X? What ‘X’ represents for in DOXIL®X?

10. We recommend that the authors to introduce polyplex micelles under section 2.2.4. Micelles. This introduction will be helpful to the scientists as the authors conceived that their review can be an introductory overview for readers. Polyplex micelles are poly(ethylene glycol) (PEG)-shielded gene delivery systems formulated upon polyionic complexation-induced self-assembly between PEG-polycation block copolymers and plasmid (p)DNA. Polyplex micelles were shown promising therapeutic outcomes in hard-to-treat cancers (Chem Rev 2018;118(14):6844-6892). For example, cyclic RGD (Arg-Gly-Asp) peptide ligand decorated polyplex micelle loading pDNA encoding human soluble form of vascular endothelial growth factor receptor-1 (or soluble fms-like tyrosine kinase-1) mediated vβ3 and vβ5 integrin-mediated uptake and showed anti-tumor activity against subcutaneously xenograftedBxPC3 human pancreatic adenocarcinoma in mice (Biomaterials 2014;35(20):5359-5368).

The authors claimed that the manuscript had been edited by a language expert. However, still, English errors were observed.  For example: A specific problem on NPs delivery to tumors, are  A specific problem in NPs delivery to tumors is

Author Response

Comments and Suggestions for Authors

The manuscript by D. Trivanovic and the team presents a fundamental review of the current status of nanoparticles in cancer management. The group discusses different types of nanoparticles ranging from organic to inorganic origin. We recommend a detailed revision addressing the following issues carefully to reach more audiences and readers of different disciplines before considering a possible publication.

  1. The authors claimed that the manuscript had been edited by a language expert. However, still, English errors were observed. For example: A specific problem on NPs delivery to tumors, are  A specific problem in NPs delivery to tumors is

We have sent the manuscript to the professional MDPI language service. We are sorry that the errors still persist. This specific point as well few other errors we managed to identify in this last reading are corrected now.

  1. Line 96: specific changes in the tumor vasculature. Please do immediately mention those specific changes. The readers will not wait for the suspense to be revealed in the next sentences about those specific changes. For example, nanosized materials accumulate in tumor tissues due to leakyvasculature and underdeveloped lymphatic drainage in the tumor region (enhanced permeability and retention effect, EPR) specific changes in the tumor vasculature.

We agree with you, and we now added more explanations as suggested. Now the line 77 starts as follows:

(…) Indeed, the property of nanosized materials to accumulate in tumor tissues due to leaky vasculature and underdeveloped lymphatic drainage. These specific changes in the tumor vasculature termed as enhanced permeability and retention effect (EPR), had been discovered decades ago by Maeda and Matsumura…(…)

  1. The prefix of dextrorotatory or levorotatory isomer should be in small capital letters. For example, Figure 2: N-acetyl-D-glucosamine  N-acetyl-d-glucosamine and D-glucosamine d-glucosamine

Follow the following instructions for small capital letters. Type small case letters l or d  Select the small case letter  Go to Format  Go to Font  Select small caps shown in “effects.”    

This is corrected now.

  1. Tumor microenvironment (TME) is a barrier in pancreatic ductal adenocarcinoma. Then the authors should mention the blood-brain barrier (BBB) in what kind of tumors. For example, barriers such as the tumor microenvironment (TME) in pancreatic ductal adenocarcinoma, or the blood–brain barrier (BBB)in brain tumors.

This is added now.

  1. The authors stated that Figure 2. Examples of polymer-based nanoparticles. The shown chemical structures are polymer-based nanoparticles? They are polymers. Not nanoparticles. Please change Examples of polymers used for preparing polymer-based nanoparticles.

This is changed accordingly.

  1. Some words were used many times throughout the manuscript. They will increase the word count but not the scientific information. Show the abbreviation or acronym only once after its appearance in the manuscript. For example, Lines 247, 268, 280, 354, 417. poly (ethylene glycol).

This is changed accordingly.

Lines 97, 364, 698 brain–blood barrier (BBB).

This is changed accordingly.

Lines 384, 417, Figure 2, poly(lactic-co-glycolic acid) (PLGA) poly(lactic-co-glycolic acid) (PLGA) and poly (lactic-co-glycolide) (PLG) are same. Why two different terminology? Use consistent terminology throughout the manuscript.

PLGA is now used in the manuscript.

  1. No special acronyms or abbreviations are required if the information is given only once. They simply increase the word count but not special scientific information.

For example, Line 103, high-intensity focused ultrasound (HIFU)

Line 416: poly (aspartic acid) (PAA),

Line 417: poly(caprolactone) PCL

Line 487: dissociation constant (KD).

Line 517: (Tf-Mpeg-pe)

Line 750: (TCGA)

Line 749: NIH

We have now removed the terms as required.

  1. Some acronyms or abbreviations were not expanded. All readers cannot immediately follow such acronyms or abbreviations.

Lines 509 and 526: HER2  human epidermal growth factor receptor 2 (HER2)

Lines 624, 626, and 648: CRISPR/cas9  clustered regularly interspersed short palindromic repeats (CRISPR)/CRISPR-associated protein 9 (Cas9)

Lines 684 and 694: IGF-1  Insulin-like growth factor 1 (IGF-1)

Line 693: Nrf2  Nuclear factor erythroid 2-related factor 2

Many more were found. Please address all of them. 

This has been adjusted now.

  1. Table 4: DOXIL®X? What ‘X’ represents for in DOXIL®X?

This is a typo error and is corrected now.

  1. We recommend that the authors to introduce polyplex micelles under section 2.2.4. Micelles. This introduction will be helpful to the scientists as the authors conceived that their review can be an introductory overview for readers. Polyplex micelles are poly(ethylene glycol) (PEG)-shielded gene delivery systems formulated upon polyionic complexation-induced self-assembly between PEG-polycation block copolymers and plasmid (p)DNA. Polyplex micelles were shown promising therapeutic outcomes in hard-to-treat cancers (Chem Rev 2018;118(14):6844-6892). For example, cyclic RGD (Arg-Gly-Asp) peptide ligand decorated polyplex micelle loading pDNA encoding human soluble form of vascular endothelial growth factor receptor-1 (or soluble fms-like tyrosine kinase-1) mediated vβ3 and vβ5 integrin-mediated uptake and showed anti-tumor activity against subcutaneously xenograftedBxPC3 human pancreatic adenocarcinoma in mice (Biomaterials 2014;35(20):5359-5368).

Thank you, we added this important information in the section 2.2.4. now along with the suggested references.